# Analysis of Variance of Multiple Causal Networks

**Zhongli Jiang**
Department of Statistics, Purdue University, West Lafayette, IN
`jiang548@purdue.edu`

**Dabao Zhang**
Department of Epidemiology & Biostatistics, University of California, Irvine, CA
dabao.zhang@uci.edu

## Abstract

Constructing a directed cyclic graph (DCG) is challenged by both algorithmic difficulty and computational burden. Comparing multiple DCGs is even more difficult, compounded by the need to identify dynamic causalities across graphs. We propose to unify multiple DCGs with a single structural model and develop a limited-information-based method to simultaneously construct multiple networks and infer their disparities, which can be visualized by appropriate correspondence analysis. The algorithm provides DCGs with robust non-asymptotic theoretical properties. It is designed with two sequential stages, each of which involves parallel computation tasks that are scalable to the network complexity. Taking advantage of high-performance clusters, our method makes it possible to evaluate the statistical significance of DCGs using the bootstrap method. We demonstrated the effectiveness of our method by applying it to synthetic and real datasets.

## 1   Introduction

Study of causal networks described by directed cyclic graphs (DCGs) becomes increasingly popular in a variety of fields, such as biomedical and social sciences [10, 3]. Detecting dynamic structures across networks sheds light on mechanistic dynamics, with promising applications. For example, gene regulatory networks related to various types of cancer differ from that of healthy cells in a complicated way and discovery of deviated regulations helps identify cancer-related biological pathways and hence design target-specific drugs [23, 18]. Interpersonal networks in social media have become a favored way for people to get informed of breaking news and natural disasters nowadays. Understanding their evolution over time may provide promising information to businesses, educators, and governments [7]. However, the complex structure and large number of nodes impose challenges on constructing even a single network, not to mention modelling and comparing multiple networks.

Structural models are widely used to describe causal networks and conduct causal inference [8, 15]. With available instrumental variables (IVs) from genomic variations, many methods have been developed to construct large causal networks of gene regulation by integrating transcriptomic and genotypic data. For instance, Liu et al. [14] proposed to assemble driver-responder relationships identified from local structural models, a formidable task due to enormous numbers of possible driver-responder pairs. Cai et al. [4] developed a sparsity-aware maximum likelihood (SML) method by putting $\ell_1$ penalty on causal effects. It may reach local maximum, leading to the identifiability issue. Chen et al. [5] proposed a two-stage penalized least square (2SPLS) approach which allows parallel computing and shows superior performance.

To infer sparse differences between two large networks, Ren and Zhang [16] recently developed a parallel algorithm based on 2SPLS and achieved better performance than separately constructing the

37th Conference on Neural Information Processing Systems (NeurIPS 2023).

two networks using 2SPLS. Zhou and Cai [25] proposed a fused sparse SEM (FSSEM) method to learn the differences through maximum likelihood inference of joint networks which is computationally infeasible with large networks. Li et al. [13] employed a Bayesian scheme in their method BFDSEM, unrealistically assuming equal sample sizes from both cohorts.

To the best of our knowledge, there is no algorithm available for investigating causal effects varying across multiple cohorts while identifying stable ones. In this paper, we develop a novel algorithm, designed for parallel computation, to construct a unified structural model for multiple causal networks. While each causal network may be depicted by a DCG, we deliver an analysis of variance (ANOVA) of these networks to identify causalities that are different across networks as well as important drivers and responders. Our algorithm is scalable in several aspects. Firstly, it is scalable to data size including sample size and number of variables. It would still be efficient and powerful under high dimension scheme. Secondly, it is scalable to the computational environment with its parallel computation. The fast computation allows to attack a daunting task in studying multiple networks, i.e., calculating $p$-values via the bootstrap method and thus controlling the false discovery rate. Thirdly, the algorithm is scalable to model complexity as it is able to infer causal networks beyond directed acyclic graphs (DAGs), such as DCGs which are demanded to depict gene regulatory networks [14].

The rest of the paper is organized as follows. We first state the model and discuss its identifiability guaranteed by available instrumental variables in Section 2. In Section 3, we present our proposed algorithm **AN**alysis **O**f **VA**riance of directed **Net**works, termed as **NetANOVA**, and introduce measures to quantify an endogenous variable's contribution as either drivers or responders. The theoretical justification is shown in Section 4 with detailed proofs in Supplemental Material. We demonstrate the feasibility and promise of our algorithm with a large-scale simulation study shown in Section 5 and a real data analysis to compare gene regulatory networks in healthy lung tissues and lung tissues with two types of cancer in Section 6. We conclude the paper with a discussion in Section 7.

## 2  A Unified Structural Model of Multiple Causal Networks

### 2.1  Model specification

We consider $K$ pertinent causal networks, each for a cohort. For each, say $k$-th, cohort we have $n^{(k)}$ observations in $(\mathbf{Y}^{(k)}, \mathbf{X}^{(k)})$, where $\mathbf{Y}^{(k)}$ is an $n^{(k)} \times p$ matrix including values of $p$ endogenous variables, and $\mathbf{X}^{(k)}$ is an $n^{(k)} \times q$ matrix including values of $q$ exogenous variables. Each variable is assumed to have mean zero. For the $k$-th causal network, we consider each, say $i$-th, endogenous variable is causally affected by other variables as follows,

$$\mathbf{Y}_i^{(k)} = \mathbf{Y}_{-i}^{(k)} \boldsymbol{\gamma}_i^{(k)} + \mathbf{X}_{\mathcal{I}_i}^{(k)} \boldsymbol{\phi}_{\mathcal{I}_i}^{(k)} + \boldsymbol{\epsilon}_i^{(k)}, \tag{1}$$

where $\mathbf{Y}_i^{(k)}$ is the $i$-th column of $\mathbf{Y}^{(k)}$, and $\mathbf{Y}_{-i}^{(k)}$ is the submatrix of $\mathbf{Y}^{(k)}$ excluding the $i$-th column. The $(p-1)$-dimensional column vector $\boldsymbol{\gamma}_i^{(k)}$ includes the direct causal effects of other endogenous variables (drivers) on the $i$-th one (responder). The set $\mathcal{I}_i$ includes the indices of exogenous variables that serve as IVs of the $i$-th endogenous variable so $\mathbf{X}_{\mathcal{I}_i}^{(k)}$ is a $n^{(k)} \times |\mathcal{I}_i|$ matrix including values of corresponding IVs. Each IV has a direct causal effect on its corresponding endogenous variable and can be identified through domain knowledge. $\boldsymbol{\phi}_{\mathcal{I}_i}^{(k)}$ includes causal effects from corresponding IVs. $\boldsymbol{\epsilon}_i^{(k)}$ includes disturbance errors which are independently distributed with mean zero and standard deviation $\sigma_i^{(k)}$.

Pooling together equation (1) for each endogenous variable, we model the $k$-th causal network as,

$$\mathbf{Y}^{(k)} = \mathbf{Y}^{(k)} \boldsymbol{\Gamma}^{(k)} + \mathbf{X}^{(k)} \boldsymbol{\Phi}^{(k)} + \boldsymbol{\epsilon}^{(k)}, \tag{2}$$

which describes a DCG as shown in Figure 1.

The $p \times p$ matrix $\boldsymbol{\Gamma}^{(k)}$ describes the causal relationships between each pair of endogenous variables and has zero diagonal elements to prohibit self-regulation. The $q \times p$ matrix $\boldsymbol{\Phi}^{(k)}$ encodes the causal effects of exogenous variables on endogenous variables. The $n^{(k)} \times p$ matrix $\boldsymbol{\epsilon}^{(k)}$ includes all disturbance errors. We assume that both $\mathbf{Y}^{(k)}$ and $\mathbf{X}^{(k)}$ have been appropriately centralized within the cohort, so no intercepts are needed in the above model.

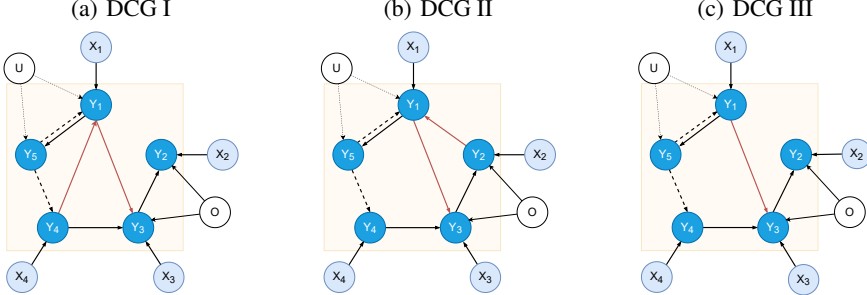

Figure 1: DCGs of three causal networks. The nodes outside the shaded regions are exogenous variables with IVs in light blue and confounding variables (unobservable $U$ and observed $O$) in white but disturbance errors are omitted. Causal relations in red differ across networks, and dashed ones cannot be revealed by available data due to unobservable $U$ or unavailable IV for $Y_5$.

## 2.2 A unified structural model

Many algorithms have been developed to construct a structural model for single causal network [14, 4, 5]. We will show that, in the interest of comparing multiple causal networks, we can also unify them into a single structural model and hence develop an appropriate algorithm to construct all networks at the same time to deliver an ANOVA of multiple networks.

Suppose we have a baseline, say $K$-th, network and are interested in others' deviation from the baseline. We first reparameterize the causal effects of all endogenous variables in (1) with

$$\boldsymbol{\beta}_i^{(k)} = \boldsymbol{\gamma}_i^{(k)} - \boldsymbol{\gamma}_i^{(K)}, \text{ for } k = 1, 2, \ldots, K-1; \quad \boldsymbol{\beta}_i^{(K)} = \boldsymbol{\gamma}_i^{(K)}. \tag{3}$$

Therefore $\boldsymbol{\beta}_i^{(K)}$ includes baseline effects and, for $k \neq K$, $\boldsymbol{\beta}_i^{(k)}$ includes deviated effects of $k$-th network from the baseline. When it is of interest to compare the networks with each other, we can similarly reparameterize to consider the deviated effects of each network from the average effects.

For any $K$ sets of $l \times m$ matrices, say $\mathbf{A}_i$ with $i = 1, 2, \cdots, K$, we define a matrix-valued function,

$$\mathcal{T}(\mathbf{A}_1, \cdots, \mathbf{A}_K) = (\mathbf{A}_{1:K}, (\text{diag}(\mathbf{A}_1^T, \cdots, \mathbf{A}_{K-1}^T), \mathbf{0}_{m(K-1)})^T),$$

where $\mathbf{A}_{1:K}$ is constructed by stocking $\mathbf{A}_1, \cdots, \mathbf{A}_K$ row-wisely, and $\mathbf{0}_{m(K-1)}$ is a $m(K-1)$-dimensional column vector with all elements zero. Further denote

$$\mathbf{Y}_i = (\mathbf{Y}_i^{(1)T}, \mathbf{Y}_i^{(2)T}, \cdots, \mathbf{Y}_i^{(K)T})^T,$$
$$\boldsymbol{\beta}_i = (\boldsymbol{\beta}_i^{(K)T}, \boldsymbol{\beta}_i^{(1)T}, \boldsymbol{\beta}_i^{(2)T}, \cdots, \boldsymbol{\beta}_i^{(K-1)T})^T,$$
$$\boldsymbol{\phi}_{\mathcal{I}_i} = (\boldsymbol{\phi}_{\mathcal{I}_i}^{(1)T}, \boldsymbol{\phi}_{\mathcal{I}_i}^{(2)T}, \cdots, \boldsymbol{\phi}_{\mathcal{I}_i}^{(K)T}),$$
$$\boldsymbol{\epsilon}_i = (\boldsymbol{\epsilon}_i^{(1)T}, \boldsymbol{\epsilon}_i^{(2)T}, \cdots, \boldsymbol{\epsilon}_i^{(K)T})^T,$$

Notice that $\boldsymbol{\gamma}_i^{(k)} = \boldsymbol{\beta}_i^{(K)} + \boldsymbol{\beta}_i^{(k)}$ for $k = 1, 2, \ldots, K-1$ from (3), we have a unified structural model for $K$ causal networks described by (2),

$$\mathbf{Y}_i = \mathcal{T}(\mathbf{Y}_{-i}^{(1)}, \mathbf{Y}_{-i}^{(2)}, \cdots, \mathbf{Y}_{-i}^{(K)})\boldsymbol{\beta}_i + \text{diag}(\mathbf{X}_{\mathcal{I}_i^{(1)}}, \mathbf{X}_{\mathcal{I}_i^{(2)}} \cdots, \mathbf{X}_{\mathcal{I}_i^{(K)}})\boldsymbol{\phi}_{\mathcal{I}_i} + \boldsymbol{\epsilon}_i. \tag{4}$$

## 2.3 Instrumental Variables, model identifiability, and confounding variables

As shown in Figure 1, the ability to reveal causal effects relies on the available IVs. An IV is an exogenous variable that only affects directly the driver but not the responder, except indirectly through its effect on the driver. This means that IVs can be used to address the model identifiability issue, which is complicated by the model's inherent endogeneity, and allow us to isolate the causal effects of endogenous variables. We specify the following assumption on available IVs following the rank condition [20], a necessary and sufficient condition for model identification.

**Assumption 1. a.** $\mathbf{X}^{(k)} \perp\!\!\!\perp \boldsymbol{\epsilon}^{(k)}$; **b.** $\exists \mathcal{C} \subset \{1, 2, \cdots, p\}$ but $\mathcal{C} \neq \emptyset$; **c.** $\forall i \in \mathcal{C}, \exists \mathcal{I}_i \subset \{1, 2, \cdots, q\}$ with $\mathcal{I}_i \neq \emptyset$ and $\boldsymbol{\phi}_{\mathcal{I}_i}^{(k)} \neq 0$; **d.** $\forall i, j \in \mathcal{C}$ with $i \neq j$, $\mathcal{I}_i \cap \mathcal{I}_j = \emptyset$; **e.** $\forall j \notin \mathcal{C}, \gamma_{ij}^{(k)} = 0$.

Assumption 1.a specifies IVs observed in $\mathbf{X}^{(k)}$ that are uncorrelated to the disturbance errors. Assumption 1.b specifies the endogenous variables in $\mathcal{C}$ that have available IVs and Assumptions 1.c and 1.d state the restrictions on the available IVs, i.e., their availability and uniqueness, respectively. While $\gamma_{ij}^{(k)}$ is the $j$-th component of $\gamma_i^{(k)}$, Assumption 1.e states that we are not able to identify any causal relation with $j$-th endogenous variable as a driver if it has no IV available, e.g., $Y_5$ in Figure 1.

A striking advantage of the IV method is its robustness to confounding but exogenous variables, which may be observable or unobservable, as shown in Figure 1. As will be demonstrated by our later algorithm, available IVs help disassociate an endogenous variable from disturbance errors of other endogenous variables in model fitting. Therefore, we may explicitly incorporate observable confounding variables to improve causal inference. However, when there are unobservable confounding variables, we can still effectively construct multiple networks with available IVs.

## 3 Model Building and Interpretation

The single structural model in (4) makes it possible to employ a limited-information approach to construct multiple causal networks even in the case of a large number of endogenous variables. Therefore we develop a two-stage algorithm NetANOVA which allows parallel computing for fast computation and hence bootstrapping the data for significance assessment. Another daunting but necessary task is to understand the multiple networks and catch the important variables in the networks and important dynamic causalities across them. We therefore propose coeffcents of determination and cause as well as correspondence analysis.

### 3.1 The algorithm NetANOVA

**The disassociation stage:** We first predict each endogenous variable in $\mathcal{C}$, solely based on all available exogenous variables, to disassociate it from the disturbance errors of other endogenous variables, following aforementioned assumption 1. For this purpose, we rearrange the terms in (2) to obtain the reduced model,

$$\mathbf{Y}^{(k)} = \mathbf{X}^{(k)}\boldsymbol{\pi}^{(k)} + \boldsymbol{\xi}^{(k)}, \tag{5}$$

where $\boldsymbol{\pi}^{(k)} = \boldsymbol{\Phi}^{(k)}(\mathbf{I} - \boldsymbol{\Gamma}^{(k)})^{-1}$ and $\boldsymbol{\xi}^{(k)} = \epsilon^{(k)}(\mathbf{I} - \boldsymbol{\Gamma}^{(k)})^{-1}$.

As shown in the real data analysis in constructing gene regulatory networks, the number of exogenous variables, i.e., the dimension $q$ for $\mathbf{X}^{(k)}$, can be much larger than the sample size $n^{(k)}$, rising issues of prediction consistency and computational time. To address them, we first apply the iterative sure independence screening (ISIS; [9]) to screen for exogenous variables and then perform regression with $\ell_2$ penalty. As later theoretical analysis shows, it allows for $q \lesssim exp(n^{(k)\theta})$ for some $\theta > 0$.

Specifically for each endogenous variable $i \in \mathcal{C}$ and $k \in \{1, \ldots, K\}$, we apply ISIS with computational complexity $O(n^{(k)}p)$ to screen for a set of exogenous variables, indexed by set $\mathcal{M}_i^{(k)}$. Using these $d = |\mathcal{M}_i^{(k)}|$ exogenous variables, we apply ridge regression to obtain the ridge estimator,

$$\hat{\boldsymbol{\pi}}_{\mathcal{M}_i^{(k)}}^{(k)} = (\mathbf{X}_{\mathcal{M}_i^{(k)}}^T \mathbf{X}_{\mathcal{M}_i^{(k)}} + \lambda_i^{(k)}I)^{-1}\mathbf{X}_{\mathcal{M}_i^{(k)}}^T \mathbf{Y}_i^{(k)} \tag{6}$$

and predict $\mathbf{Y}_i^{(k)}$ with

$$\hat{\mathbf{Y}}_i^{(k)} = \mathbf{X}_{\mathcal{M}_i^{(k)}}^{(k)} \hat{\boldsymbol{\pi}}_{\mathcal{M}_i^{(k)}}^{(k)}, \tag{7}$$

where $\lambda_i^{(k)}$ is a tuning parameter that can be selected via generalized cross validation [11].

**The inference stage:** We use the predicted endogenous variables in the previous stage to identify and estimate the causal effects, hence constructing the causal networks. We first calculate the projection matrix,

$$\mathbf{P}_i^{(k)} = \mathbf{I}_{n^{(k)}} - \mathbf{X}_{\mathcal{I}_i}^{(k)} \left( \mathbf{X}_{\mathcal{I}_i}^{(k)T} \mathbf{X}_{\mathcal{I}_i}^{(k)} \right)^{-1} \mathbf{X}_{\mathcal{I}_i}^{(k)T}.$$

Multiplying the projection matrix $\mathbf{P}_i = \text{diag}\{\mathbf{P}_i^{(1)}, \mathbf{P}_i^{(2)}, \ldots, \mathbf{P}_i^{(K)}\}$ to both sides of model (4), we can eliminate the exogenous variables from the model and get,

$$\mathbf{P}_i\mathbf{Y}_i = \mathbf{P}_i\boldsymbol{\Upsilon}_{-i}\boldsymbol{\beta}_i + \mathbf{P}_i\boldsymbol{\epsilon}_i, \tag{8}$$

where $\boldsymbol{\Upsilon}_{-i} = \mathcal{T}(\mathbf{Y}_{-i}^{(1)}, \mathbf{Y}_{-i}^{(2)}, \cdots, \mathbf{Y}_{-i}^{(K)})$. Since $\mathbf{P}_i \boldsymbol{\Upsilon}_{-i}$ and $\boldsymbol{\epsilon}_i$ are correlated, we instead regress $\mathbf{P}_i \mathbf{Y}_i$ against $\mathbf{P}_i \hat{\boldsymbol{\Upsilon}}_{-i}$ with $\hat{\boldsymbol{\Upsilon}}_{-i} = \mathcal{T}(\hat{\mathbf{Y}}_{-i}^{(1)}, \hat{\mathbf{Y}}_{-i}^{(2)}, \cdots, \hat{\mathbf{Y}}_{-i}^{(K)})$ which is disassociated from $\boldsymbol{\epsilon}_i$. With possibly high-dimensional $\boldsymbol{\beta}_i$, we apply adaptive lasso [26] to obtain its estimator,

$$\hat{\boldsymbol{\beta}}_i = \arg\min_{\boldsymbol{\beta}_i} \left\{ \frac{1}{n} ||\mathbf{P}_i \mathbf{Y}_i - \mathbf{P}_i \hat{\boldsymbol{\Upsilon}}_{-i} \boldsymbol{\beta}_i||_2^2 + \nu_i \hat{\boldsymbol{\omega}}_i^T |\boldsymbol{\beta}_i|_1 \right\},$$

where $\nu_i$ is a tuning parameter, and $\hat{\boldsymbol{\omega}}_i = |\hat{\boldsymbol{\beta}}_{0i}|^{-\gamma}$ for some $\gamma > 0$ where $|\hat{\boldsymbol{\beta}}_{0i}|_1$ contains the absolute values of $\hat{\boldsymbol{\beta}}_{0i}$, which is a preliminary estimator. The original networks can be recovered by calculating $\hat{\boldsymbol{\gamma}}_i^{(k)} = \hat{\boldsymbol{\beta}}_i^{(K)} + \hat{\boldsymbol{\beta}}_i^{(k)}$ for $k = 1, 2, \ldots, K-1$.

In summary of these two stages, we summarize the algorithm in Algorithm 1.

---

**Algorithm 1 ANalysis Of VAriance of directed Networks (NetANOVA)**

---

**Input:** $(\mathbf{Y}^{(k)}, \mathbf{X}^{(k)})$, $k \in \{1, 2, ..., K\}$, with each variable centralized within cohort but scaled according to the baseline cohort; Predefined index set $\mathcal{I}_i$ for each $i \in \mathcal{C}$; $d \leftarrow O(n_{\min}^{1-\theta})$.
**STAGE 1**
**for** $i \in \mathcal{C}$ **do**
    1. Reduce the dimension of $\mathbf{X}^{(k)}$ by ISIS to get $\mathbf{X}_{\mathcal{M}_i^{(k)}}^{(k)}$; Set $\mathbf{X}_{\mathcal{M}_i^{(k)}}^{(k)} = \mathbf{X}^{(k)}$ if $q \leq n^{(k)}$.

    2. Estimate $\hat{\mathbf{Y}}_i^{(k)}$ by regressing $\mathbf{Y}_i^{(k)}$ against $\mathbf{X}_{\mathcal{M}_i^{(k)}}^{(k)}$ with ridge regression.

**end for**
**STAGE 2**
**for** $i = 1, 2, \ldots, p$ **do**
    1. Calculate projection matrices $\mathbf{P}_i$.
    2. Predict $\hat{\boldsymbol{\Upsilon}}_{-i}$ from $\hat{\mathbf{Y}}_i^{(k)}$, $i \in \mathcal{C}$.
    3. Estimate $\hat{\boldsymbol{\beta}}_i$ by regressing $\mathbf{P}_i \mathbf{Y}_i$ against $\mathbf{P}_i \hat{\boldsymbol{\Upsilon}}_{-i}$ with adaptive lasso.
**end for**
**Output:** $\hat{\boldsymbol{\beta}}_1, \ldots, \hat{\boldsymbol{\beta}}_p$ which contains the baseline and differential causal effects.

---

### 3.2 Coefficient of determination and coefficient of cause

For each endogenous variable $i$, we can calculate its coefficient determination to measure the proportion of its variation due to the effects of its drivers, i.e., $\mathbf{Y}_{-i}^{(k)} \boldsymbol{\gamma}_i^{(k)}$ in (1), for cohort $k$,

$$R_i^{2(k)} = 1 - ||\mathbf{Y}_i^{(k)} - \mathbf{Y}_{-i}^{(k)} \hat{\gamma}_i^{(k)}||_2^2 / ||\mathbf{Y}_i^{(k)}||_2^2,$$

On the other hand, we can also calculate the coefficient of cause for each endogenous variable $i$ which summarizes proportions that it contributes to the variation of its responders, for cohort $k$,

$$C_i^{2(k)} = \sum_{j=1}^{p} \left( 1 - ||\mathbf{Y}_j^{(k)} - \mathbf{Y}_i^{(k)} \hat{\gamma}_{ji}^{(k)}||_2^2 / ||\mathbf{Y}_j^{(k)}||_2^2 \right).$$

While we name the above coefficient of determination as causal $R^2$ with its value between zero and one, we will simply call the coefficient of cause $C^2$ which is positive but may be greater than one.

### 3.3 Correspondence analysis of causal effects

We can conduct a correspondence analysis of causal effects in $\boldsymbol{\Gamma}^{(k)}$ to reveal clusters of drivers, responders, or driver-responder pairs which outstand from the rest in each cohort $k$. We can also conduct a correspondence analysis of deviated causal effects to reveal these clusters which deviate the most from a baseline or the rest. However, each causal effect or its deviation may vary differently and we need to standardize them, based on the bootstrap results, before correspondence analysis. For example, to compare cohorts $k$ and $l$ for their causal effects, we should instead obtain the following

summary statistic,

$$z_{ij}^{(k,l)} = \left( \hat{\gamma}_{ij}^{(k)} - \hat{\gamma}_{ij}^{(l)} \right) \Big/ \left( \sum_{b=1}^{B} (\hat{\gamma}_{ij}^{(k,b)} - \hat{\gamma}_{ij}^{(l,b)})^2 / (B-1) \right)^{1/2}.$$

where $\hat{\gamma}_{ij}^{(k)}$ and $\hat{\gamma}_{ij}^{(k,b)}$ are estimates of corresponding effects from the observed data and $b$-th set of bootstrap data, respectively; we also let $z_{ij}^{(k)} = 0$ when both of its numerator and denominator are sufficiently small. We can define similar statistics to investigate effects' variation from their means.

A singular value decomposition (SVD) of $Z^{(k,l)} = (z_{ij}^{(k,l)})_{p \times p}$ can obtain its left and right singular vectors, say $\{U_i^{(k,l)}\}$ and $\{V_i^{(k,l)}\}$, respectively. While the left singular vectors help identify responders that differ between the two cohorts, the right singular vectors help identify drivers that differ between the two cohorts. We can overlay the plot of $U_2^{(k)}$ vs. $U_1^{(k)}$ with the plot of $V_2^{(k)}$ vs. $V_1^{(k)}$ to identify responder-driver pairs for their causalities varying the most across networks.

## 4 Theoretical Analysis

In this section, we will establish non-asymptotic guarantees and show that our constructed DCGs have good theoretical properties. We characterize the properties with a prespecified sequence $\delta^{(k)} = \exp\{-o(n^{(k)})\}$ with $\delta^{(k)} \to 0$ as $n^{(k)} \to \infty$, i.e., each $\delta^{(k)}$ approaches zero slower than $\exp\{-n^{(k)}\}$. We denote $\delta_{min} = \min_{1 \leq k \leq K} \delta^{(k)}$ and only present main results here, with details including general notations, assumptions, and proofs in Supplementary Material.

**Theorem 4.1.** *Let* $n = \sum_{k=1}^{K} n^{(k)}$, $\mathcal{S}_i = supp(\boldsymbol{\beta}_i)$, *and* $g_n$ *is a function of* $n$ *and* $\delta_{min}$ *specified in (9) in Supplemental Material. Then under Assumption 1 and Assumptions B.1–B.4 in Supplementary Material, we have that, with probability at least* $1 - \delta_{min} - \delta$ *with* $\delta = p \sum_{k=1}^{K} \delta^{(k)}$,

1. *(Bounded Errors)* $\|\hat{\boldsymbol{\beta}}_i - \boldsymbol{\beta}_i\|_2^2 \lesssim |S_i| \, g_n \, \{d \vee \log(pK/\delta) \vee \|\boldsymbol{\pi}\|_{2max}^2 \vee \log(d/\delta_{min})\}/n;$

2. *(Causality Consistency)* $sign(\hat{\boldsymbol{\beta}}_i) = sign(\boldsymbol{\beta}_i)$.

We will next show that, with proper choice of $\{\delta^{(k)}\}$, we can bound the estimation errors system-wise with ultra high probability. We can pick $\{\delta^{(k)}\}$ such that $\delta_{min} \asymp e^{-n^t}$ where $t \in (0, min(\theta, 1-\theta))$. Then we have $f_n \lesssim n^{(2-\theta)/2}$, and the restriction on the number of true signals can be reduced to $|\mathcal{S}_i| \lesssim n^{\theta/2}$, which is a requirement that can be fulfilled especially in a sparse model. We have $d = n^{1-\theta}$, so $g_n \lesssim n^{(1-2\theta)/2}$. As a result, the bound can be dominated by the order of $|S_i|n^{(1-4\theta)/2}$.

Note that the error for the whole system of $p$ nodes can be controlled by a similar bound, replacing each occurrence by $|S_i|$ with $\max_{1 \leq i \leq p} |S_i|$, with probability at least $1 - p(\delta_{min} + \delta)$. Following the former calculation, we learn that, when $\max_{1 \leq i \leq p} |S_i|$ grow slower than $n^{(4\theta-1)/2}$, the error will approach zero when $n \to \infty$. To achieve the bound with high probability, we only need $p\delta$ to diminish. That is, we could have the error well controlled even if the dimension grows up to $e^{n^s}$ where $s \in (0, \frac{t}{2})$.

Given the auspicious estimation performance, we will further discuss the benefit of pooling cohorts compared to that of learning in a single-cohort analysis, where the estimators $\beta_i^{(k)}$ for each cohort are estimated separately using the algorithm without re-parameterization. For each $k$, we can separately conduct the analysis on each cohort and obtain the following property for single task estimation.

**Corollary 4.1.1.** *Under the same conditions as Theorem 4.1, we have that, with probability at least* $1 - \delta_{min} - \delta/K$,

$$\|\hat{\boldsymbol{\gamma}}_i^{(k)} - \boldsymbol{\gamma}_i^{(k)}\|_2^2 \lesssim |S_i| \, g_n \, \{d \vee \log(pK/\delta) \vee \|\boldsymbol{\pi}\|_{2max}^2 \vee \log(d/\delta_{min})\}/n^{(k)}.$$

The proof directly follows by treating each cohort as a baseline cohort in Theorem 4.1. Note that the denominator is making a crucial difference in inferring the bound. The algorithm indicates that when we aggregate the samples that does not differ too much, the estimator tends to converge faster,

especially when $n \gg n^{(k)}$, where we have a plethora of samples in addition to the samples for the base cohort.

With the same choice of $\delta_{min}$ and $\delta$ as the above theorem, the probability for consistent variable selection can approach one, with a less stringent requirement in the size of $S_i$. The above theorem implies that our proposed method can identify both baseline and differential regulatory effects, among all of the $K$ networks with a sufficiently large probability, not only in terms of the set of true signals but also the sign of signals. In the case of gene regulatory networks, for instance, our method could correctly distinguish the up and down regulations between genes.

Next, we will present a theorem for the coefficient of determination and coefficient of cause defined in Section 3.2. Denote the statistics $R^2$ and $C^2$ calculated with real parameters as, respectively,

$$R_{0i}^{2(k)} = 1 - ||\mathbf{Y}_i^{(k)} - \mathbf{Y}_{-i}^{(k)}\gamma_i^{(k)}||_2^2/||\mathbf{Y}_i^{(k)}||_2^2, \quad C_{0i}^{2(k)} = \sum_{j=1}^{p}(1 - ||\mathbf{Y}_j^{(k)} - \mathbf{Y}_i^{(k)}\gamma_{ji}^{(k)}||_2^2/||\mathbf{Y}_j^{(k)}||_2^2).$$

Then we can derive the following properties.

**Theorem 4.2.** *(Coefficients of Determination/Cause Consistency) Under the same conditions as in Theorem 4.1, with $h_n = \sqrt{n^{(k)}} + \sqrt{-\log(\delta_{min})} + 1$, we have that,*

1. *With probability at least $1 - p(2\delta_{min} + \delta/K)$,*

$$\sum_{i=1}^{p}|R_i^{2(k)} - R_{0i}^{2(k)}| \quad \lesssim \quad \frac{|\mathcal{S}_i|g_n\{d \vee \log(p/\delta) \vee ||\boldsymbol{\pi}||_{2max}^2 \vee \log(d/\delta_{min})\}}{n^{(k)3/4}}(1 + ||\phi_{\mathcal{I}_i}^{(k)}||_2 h_n);$$

2. *With probability at least $1 - p(\delta_{min} + \delta/K)$,*

$$\sum_{i=1}^{p}|C_i^{2(k)} - C_{0i}^{2(k)}| \quad \lesssim \quad \frac{|\mathcal{S}_i|g_n\{d \vee \log(p/\delta) \vee ||\boldsymbol{\pi}||_{2max}^2 \vee \log(d/\delta_{min})\}}{n^{(k)}}(1 + ||B||_1 \vee 1).$$

As considered in Theorem 4.1, we set $\delta_{min} \asymp \delta/(pK) \asymp e^{-n^t}$. Then $h_n \lesssim \sqrt{n^{(k)}}$, and $\sum_{i=1}^{p}|R_i^{2(k)} - R_{0i}^{2(k)}| \lesssim \max_{1 \leq i \leq p}|S_i|n^{(1-4\theta)/2} + (\max_{1 \leq i \leq p}|S_i|)^{1/2}n^{\frac{-2\theta-1}{2}}$ which would vanish if $\max_{1 \leq i \leq p}|S_i| \lesssim n^{(4\theta-1)/2}$. Similarly, $\sum_{i=1}^{p}|C_i^{2(k)} - C_{0i}^{2(k)}| \lesssim \max_{1 \leq i \leq p}|S_i|n^{(1-4\theta)/4}$ and will converge to 0 when $\max_{1 \leq i \leq p}|S_i| \lesssim n^{(4\theta-1)/4}$. From the above discussion, both $R^2, C^2$ for all the nodes in the whole system is $\ell_1$ consistent. It is worth noting that, although our algorithm fits a model for each responder, estimate of each node's explanatory power as a driver is also well controlled over the system, but with a slightly larger bound as shown in the above theorem.

## 5   Simulation Study

We examine the performance of NetANOVA by simulating data of sample size 200, 500 and 1000 from each of three pertinent DCGs, consisting of 1000 endogenous variables. However, only 50 endogenous variables are involved with causality, with each regulated by 3 others on average. Each of the three networks has 5 unique causal effects. First two networks share five causal effects but with opposite signs to that of the third one, resulting in the first two networks having 15 different effects from the third one and the first network have 20 different effects from the second one. The size for causal effects is taken from a uniform distribution over $[-0.8, -0.3] \bigcup [0.3, 0.8]$. The IVs are generated from a multinomial distribution with 3 outcomes 0, 1, 2, with probabilities 0.25, 0.5, 0.25 respectively. The disturbance errors are sampled independently from $N(0, 0.1^2)$.

For each sample size, we simulated 100 data sets and applied NetANOVA to each by bootstrapping 100 times to calculate $p$ values and construct Receiver Operating Characteristic (ROC) curves as shown in Figure 2. For causal effects of all three DCGs, Figure 2.a shows an almost perfect area under curve (AUC) for each sample size. When comparing causal effects of the first two DCGs vs. the baseline one, i.e., the third DCG, the AUC bottoms at 0.867 with sample size at 200, but reaches one with sample size at 1000, showing excellent performance of NetANOVA.

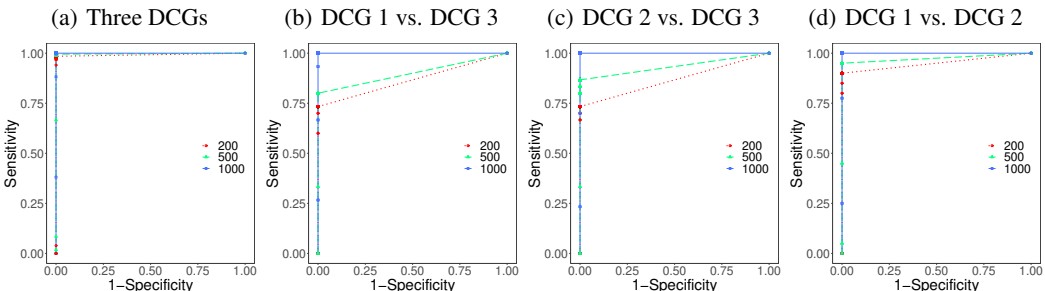

Figure 2: ROC curves of **NetANOVA** on simulated data. (a) ROC curves of causal effects constructed in all three networks; (b) ROC curves of deviated effects of the first network from the third one; (c) ROC curves of deviated effects of the second network from the third one; (d) ROC curves of deviated effects of the first network from the second one.

Taking one simulated dataset with sample size at 500, we evaluated our proposed statistics $R^2$ and $C^2$ by plotting estimated statistics against the statistics calculated with true causal effects in Figure 3. For each of the three networks, we observe linear trends with slope almost one for both $R^2$ and $C^2$ statistics, indicating their excellent performance.

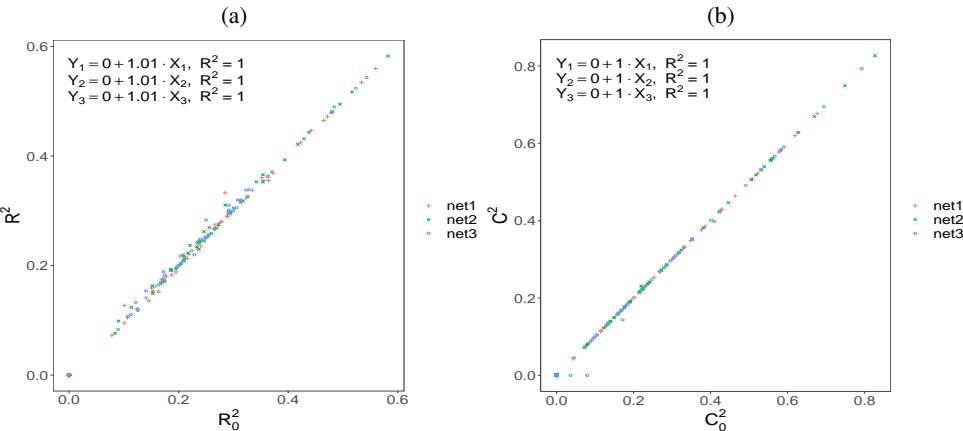

Figure 3: Statistics $R^2$ and $C^2$ on a simulated data with sample size 500. We plot each statistic ($R^2$ or $C^2$) vs. its value calculated with true causal effects ($R_0^2$ or $C_0^2$) for each network. We simply label the three networks as 1, 2, and 3. Shown in the top-left corners are the fitted linear regression models between the pairs as well as the corresponding coefficients of determination.

We also took one simulated data with sample size 500 and conducted our proposed correspondence analysis to identify drivers and responders that show important dynamic causality across the three DCGs. When comparing DCG I vs. DCG III, Figure 4.b shows a driver-responder pair (14, 45) with different causal effects, which is verified by observing 14 significantly up regulates 45 in DCG I vs. DCG III in Figure 4.a (blue connections). However, we also see the driver 18 and responder 45 stay opposite on the $y$-axis, because 18 significantly down regulates 45 implying in DCG I vs. DCG III in Figure 4.a. On $x$-axis, we see pairs (12,13) and (12,7) which correspond to another cluster of causal networks as shown in the right of Figure 4.a.

## 6   Real Data Analysis

We applied NetANOVA to investigate the gene regulatory networks of lung tissues of healthy individuals and patients with lung adenocarcinoma (LUAD) or lung squamous cell carcinoma (LUSC). We obtained transcriptomic and genotypic data of healthy lungs ($n = 482$) from the Genotype-Tissue Expression (GTEx) project [6] and of both LUAD ($n = 485$) and LUSC ($n = 406$) from The Cancer

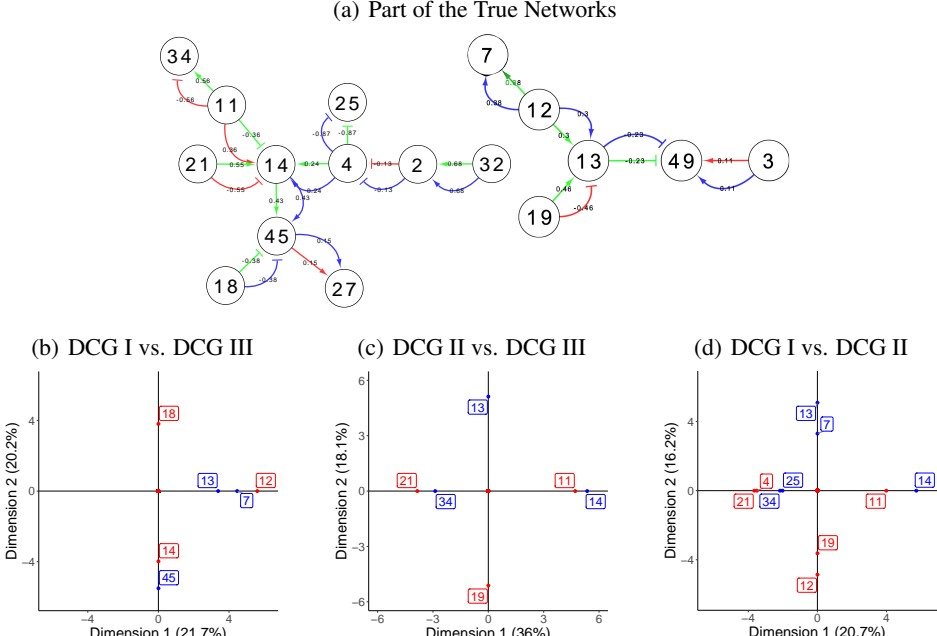

Figure 4: Correspondence analysis of **NetANOVA** on a simulated data with sample size 500. (a) Plot of some deviated causal effects among the three true networks (shown in Figure 6) with DCG I vs. DCG III, DCG II vs. DCG III, and DCG I vs. DCG II in blue, red, and green, respectively. There are a total of 1000 endogenous variables with the first 50 involved in causal relations. (b)-(d) Plots for correspondence analysis with drivers in red and responders in blue.

Genome Atlas (TCGA) project [22]. After pre-processing, there are 15,135 genes and 427,820 single nucleotide polymorphisms (SNPs) being shared by three cohorts. Cis-eQTL mapping identified 7059 genes with at least one significant SNPs inside their genetic regions (p-value$< 0.05$), i.e., valid IVs. We bootstrapped the data 100 times to assess the significance of all effects with results shown in Table 1.

Table 1: Summary of causal effects identified by **NetANOVA**. Shown in the columns are the results from the original data, different bootstrap cutoffs (80% - 100%), and adjusted by Benjamini-Hochberg adjustment (BH), respectively.

| Type of Effects | Original | 80% | 90% | 95% | 100% | BH |
|---|---|---|---|---|---|---|
| Healthy Tissue | 79833 | 16594 | 11481 | 8760 | 4602 | 3165 |
| LUAD vs Healthy Tissue | 13711 | 1185 | 848 | 670 | 458 | 296 |
| LUSC vs Healthy Tissue | 13104 | 1385 | 980 | 768 | 477 | 274 |
| LUSC vs LUAD | 18139 | 1615 | 976 | 665 | 289 | 38 |

Controlling adjusted $p$-value at 0.1, we have the largest subnetwork bearing differential structures shown in Figure 5.a, which is verified via STRING [21] as in Figure 5.b. STRING reports a protein-protein interaction (PPI) enrichment with $p$-value $< 10^{-16}$, implying significant causal effects between the genes shown in Figure 5.a. We identified many previous validated relationships, such as the connected pair RPS14 and RACK1 (GNB2L1) which were experimentally verified by [1]. Our results suggest new findings, particularly causal relationship rather than mere association, e.g., deviated regulations between ARL10 and CLTB from healthy lung tissues in both LUAD and LUSC. Our correspondence analysis of these deviated effects, shown in Figure 5.c-e, confirms these important driver-responder pairs, and the calculated statistics $R^2$ and $C^2$ in Table 2 also show such deviation between the three cohorts.

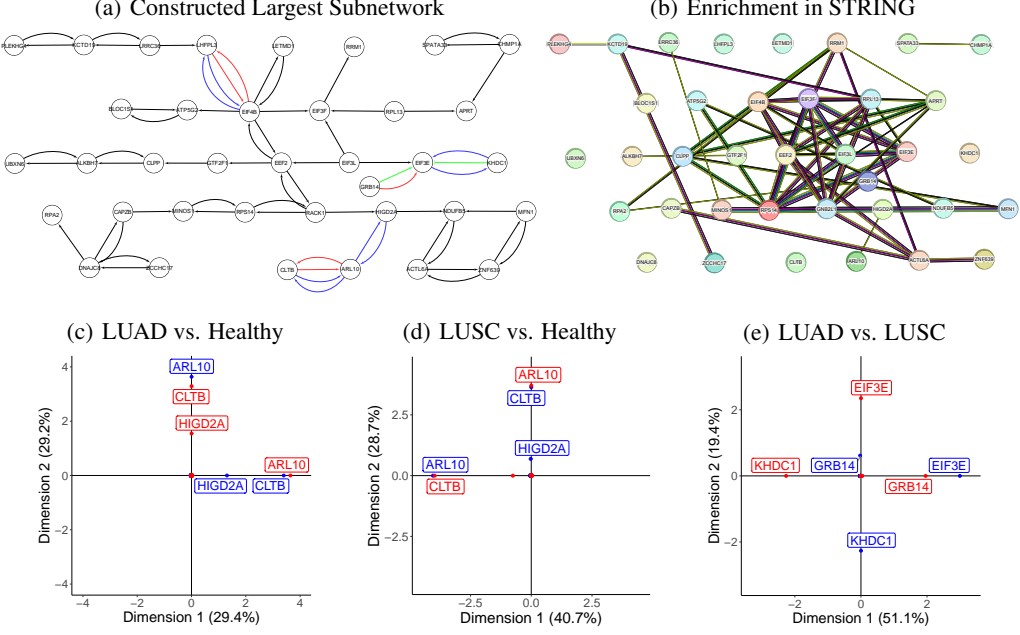

Figure 5: Partial results of gene regulatory networks for human lung. (a) The largest subnetworks of gene regulation with deviated causal effects between healthy, LUAD, and LUSC lung tissues. Shown in black is the gene regulatory network for healthy lung tissue. Shown in blue, red, green are significantly deviated causal effects of LUAD vs. healthy, LUSC vs. healthy, and LUAD vs. LUSC, respectively. (b) Enrichment of the largest subnetwork in STRING. (c)-(e) Correspondence analysis of the deviated effects between the largest subnetworks with drivers in red and responders in blue.

## 7  Discussion

Though having promising applications, construction and dynamic study of multiple causal networks are computationally challenging in practice due to the involved large systems. We develop the algorithm **NetANOVA** which avoids optimizing full-information objective functions of the whole system and instead takes limited-information objective functions, each for revealing all drivers of a single responder. Such a responder-focused method allows to deploy parallel computation in two sequential stages and makes it possible to be computationally scalable to the number of involved responders. With available clusters of computers, we can further take the bootstrap method for the usually infeasible task, i.e., evaluating the significance of the constructed causalities in each network and dynamic causalities across networks. Our theoretical analysis and simulation study demonstrate the utility and efficiency of NetANOVA.

Table 2: Statistics $R^2$ and $C^2$ for Important Genes

|       | Cohort  | ARL10 | CLTB | EIF3E | GRB14 | HIGD2A | KHDC1 |
|-------|---------|-------|------|-------|-------|--------|-------|
|       | LUAD    | 0.90  | 0.74 | 0.73  | 0.58  | 0.71   | 0.60  |
| $R^2$ | LUSC    | 0.79  | 0.79 | 0.77  | 0.75  | 0.54   | 0.00  |
|       | Healthy | 0.00  | 0.00 | 0.56  | 0.00  | 0.30   | 0.00  |
|       | LUAD    | 1.41  | 0.74 | 1.18  | 0.00  | 0.86   | 0.60  |
| $C^2$ | LUSC    | 0.79  | 0.79 | 0.75  | 0.75  | 0.16   | 0.00  |
|       | Healthy | 0.00  | 0.00 | 0.00  | 0.00  | 0.34   | 0.00  |

With the model complexity of multiple causal networks, especially DCGs that NetANOVA supports, it is a daunting task to visualize them and comprehend each network and variation across multiple networks. We have proposed statistics $R^2$ and $C^2$ to quantify the contributions of responders and drivers within a causal network, respectively. While comparing these two statistics across networks may help identify responders and drivers involved in network dynamics, we develop a correspondence analysis to visualize the key players. While our simulation study and real data analysis show promising results, correspondence analysis could be further developed to realize its full potential.

## Acknowledgments and Disclosure of Funding

This research is partially supported by the Office of Research of Purdue University and the National Cancer Institute under Grant R03 CA235363. The results shown here are in whole or part based upon data generated by the TCGA Research Network: https://www.cancer.gov/tcga. The Genotype-Tissue Expression (GTEx) Project was supported by the Common Fund of the Office of the Director of the National Institutes of Health, and by NCI, NHGRI, NHLBI, NIDA, NIMH, and NINDS. The data used for the analyses described in this manuscript were obtained from the GTEx Portal on 10/03/2021 and dbGaP accession number phs000424.v8.p2 on 7/23/2019.

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
