# A List of Notations

**For a matrix A:**

| | |
|---|---|
| $\mathbf{A}_{ij}$ | Entry of the $i$-th row and the $j$-th column of the matrix |
| $\lVert\mathbf{A}\rVert_1$ | Maximum column sums of the absolute value of the matrix. |
| $\lVert\mathbf{A}\rVert_{-\infty}$ | Minimum of the row sums of absolute values of the matrix. |
| $\lVert\mathbf{A}\rVert_{\infty}$ | Maximum of the row sums of absolute values of the matrix. |
| $\lambda_{\max}(\mathbf{A})$ | Maximum eigenvalue of a matrix. |
| $\lambda_{\min}(\mathbf{A})$ | Minimum eigenvalue of a matrix. |

**For a vector $\nu$ :**

| | |
|---|---|
| $\boldsymbol{\nu}_i$ | $i$-th element of $\boldsymbol{\nu}$. |
| $\lVert\boldsymbol{\nu}\rVert_q$ | $\left(\sum_{i=1}^n \lvert\boldsymbol{\nu}_i\rvert^q\right)^{1/q}$ for $q = 1, 2$. |

**For numbers $x$ and $y$:**

| | |
|---|---|
| $x \wedge y$ | Minimum of numbers $x$ and $y$. |
| $x \vee y$ | Maximum of numbers $x$ and $y$. |
| $x \lesssim y$ | $x \le cy$ for some positive constant $c$. |
| $x \gtrsim y$ | $x \ge cy$ for some positive constant $c$. |
| $x \asymp y$ | $cx \le y \le dx$ for some positive constant $c, d$. |

# B Detailed Theoretical Analysis

In this section, we will present detailed assumptions besides Assumption 1 and systematically analyze theoretical properties of our proposed NetANOVA algorithm, which lay the groundwork for proving the theorems stated in the main text.

Our theoretical analysis is conducted, assuming a prespecified sequence $\delta^{(k)} = \exp\{-o(n^{(k)})\}$ with $\delta^{(k)} \to 0$ as $n^{(k)} \to \infty$, i.e., each $\delta^{(k)}$ approaches zero slower than $\exp\{-n^{(k)}\}$. We also denote

$$\delta_{min} = \min_{1 \le k \le K} \delta^{(k)}.$$

## B.1 General Results

To prove the consistency of the disassociation stage, we first point out that $\mathcal{M}_i^{(k)}$ obtained from ISIS [24] successfully recovers the true nonzero set $\mathcal{M}_{i*}^{(k)}$ in reduced form (5). We first state the assumption which restricts the sample size for different cohorts in the same order and ensures the sparsity of the true underlying relationships. We state the assumption by extending the conditions in Fan and Lv [9] to pave the way for Theorem B.1, the sure screening property.

Denote

$$\Sigma^{(k)} = \mathrm{Cov}(\mathbf{X}^{(k)}), \quad W^{(k)} = (\Sigma^{(k)})^{-1/2}\mathbf{X}^{(k)T},$$

and, for any index subset $\mathcal{M} \subset \{1, 2, \cdots, q\}$,

$$\Sigma_{\mathcal{M}}^{(k)} = cov(\mathbf{X}_{\mathcal{M}}^{(k)}), \quad W_{\mathcal{M}}^{(k)} = (\Sigma_{\mathcal{M}}^{(k)})^{-1/2}\mathbf{X}_{\mathcal{M}}^{(k)T}$$

Further denote the $j$-th row of $\boldsymbol{Y}_i^{(k)}$, $\boldsymbol{X}_i^{(k)}$, and $\boldsymbol{\pi}_i^{(k)}$ as $Y_{ji}^{(k)}$, $X_{ji}^{(k)}$, and $\pi_{ji}^{(k)}$, respectively.

**Assumption B.1.**

*(a)* $\lambda_{max}(\Sigma^{(k)}) \lesssim (n^{(k)})^{\tau^{(k)}}$ *for some positive* $\tau^{(k)}$.

*(b)* $W^{(k)}$ *follows a spherically symmetric distribution with concentration property: There exist some constants* $\tilde{c}_1^{(k)} > 1$, $\tilde{c}_2^{(k)} > 1$, *and* $\tilde{c}_3^{(k)} > 0$ *such that, for any index subset* $\mathcal{M} \subset \{1, 2, \cdots, q\}$ *with* $\lvert\mathcal{M}\rvert \ge \tilde{c}_1^{(k)} n^{(k)}$, *we have, with probability at least* $1 - \exp(-\tilde{c}_3^{(k)} n^{(k)})$,

$$1/\tilde{c}_2^{(k)} \le \lambda_{min}(W_M^{(k)T} W_M^{(k)}/\lvert\mathcal{M}\rvert) \le \lambda_{max}(W_M^{(k)T} W_M^{(k)}/\lvert\mathcal{M}\rvert) \le \tilde{c}_2^{(k)}.$$

*(c)* $var(Y_{ji}^{(k)}) \lesssim 1$ *and there exists* $\kappa^{(k)} \geq 0$ *such that*

$$\min_{m \in \mathcal{M}_{i*}^{(k)}} \left| \pi_{mi}^{(k)} \right| \gtrsim (n^{(k)})^{-\kappa^{(k)}} \quad \text{and} \quad \min_{m \in \mathcal{M}_{i*}^{(k)}} \left| cov\left( Y_{ji}^{(k)}, X_{jm}^{(k)} \pi_{mi}^{(k)} \right) \right| \gtrsim 1.$$

For each node $i$ in network $k \in \{1, 2, ..., K\}$, we have the following theorem.

**Theorem B.1.** *Denote* $U^{(k)} = 1 - 2\kappa^{(k)} - \tau^{(k)}$, $U_{min} = \min_{1 \leq k \leq K} U^{(k)}$, $\kappa_{max} = \max_{1 \leq k \leq K} \kappa^{(k)}$, *and* $q \lesssim \exp(n_{min}^{(k)})^{\tilde{c}}$ *for some* $\tilde{c} \in (0, 1 - 2\kappa^{(k)})$ *where* $n_{\min} = \min_{1 \leq k \leq K} n^{(k)}$, *then under Assumption B.1, there exists some* $\theta \in (0, U_{min})$ *and some positive constant c, such that, with probability at least* $1 - c \exp\left\{ -(n^{(k)})^{1 - 2\kappa_{max}} / \log(n^{(k)}) \right\}$,

$$\mathcal{M}_{i*}^{(k)} \subseteq \mathcal{M}_i^{(k)}.$$

Theorem B.1 implies that ISIS can recover the true nonzero set $\mathcal{M}_{i*}^{(k)}$ for each network with overwhelming probability, when $n$ tends to infinity, and thus paves the way for our subsequent analysis. We will establish the rest of the theory based on $\mathcal{M}_{i*}^{(k)} \subseteq \mathcal{M}_i^{(k)}$ for convenience.

Next, we will investigate the consistency of predictions using ridge regression in (7). To facilitate simplicity, we will exclude the subscript $\mathcal{M}_i^{(k)}$ from $\mathbf{X}_{\mathcal{M}_i^{(k)}}^{(k)}$ and replace it with $\mathbf{X}_i^{(k)}$ from now on.

Similarly, we will refer to the true and estimated causal parameters as $\boldsymbol{\pi}_i^{(k)}$ and $\hat{\boldsymbol{\pi}}_i^{(k)}$, respectively.

**Assumption B.2.** $\lambda_{max}(\mathbf{X}_i^{(k)T} \mathbf{X}_i^{(k)}) \asymp \lambda_{min}(\mathbf{X}_i^{(k)T} \mathbf{X}_i^{(k)}) \asymp n^{(k)}$ *and the singular values of* $\mathbf{I} - \boldsymbol{\Gamma}^{(k)}$ *have a positive lower bound for each* $k \in 1, \ldots, K$.

Let

$$\boldsymbol{\Upsilon} = \mathcal{T}(\mathbf{Y}^{(1)}, \mathbf{Y}^{(2)}, \cdots, \mathbf{Y}^{(K)}), \quad \boldsymbol{\Pi} = \mathcal{T}(\boldsymbol{\pi}^{(1)}, \boldsymbol{\pi}^{(2)}, \cdots, \boldsymbol{\pi}^{(K)}),$$

and denote

$$\mathbf{X} = \text{diag}\{\mathbf{X}_i^{(1)}, \mathbf{X}_i^{(2)}, \ldots, \mathbf{X}_i^{(K)}\}.$$

We use $\hat{\boldsymbol{\Upsilon}}$ and $\hat{\boldsymbol{\Pi}}$ to denote the prediction of $\boldsymbol{\Upsilon}$ and $\boldsymbol{\Pi}$, respectively. In addition, we will use the subscript $j$ to denote the $j$-th column of the corresponding matrices. We further denote

$$||\boldsymbol{\pi}||_{2max}^2 = \max_{1 \leq i \leq p} \{||\boldsymbol{\pi}_i^{(1)}||_2^2 \vee ||\boldsymbol{\pi}_i^{(2)}||_2^2 \vee \cdots \vee ||\boldsymbol{\pi}_i^{(K)}||_2^2\},$$

which represents the maximum of the square of the magnitude of signals over all networks and nodes. We have the following theorem for all $j \in \{1, 2, \ldots, Kp\}$.

**Theorem B.2.** *Let the ridge parameter in (6) be, for each node $i$,* $\lambda_i^{(k)} = \sqrt{(n^{(k)})}$. *Under Assumptions B.1 and B.2, we have, with probability at least* $1 - \sum_{k=1}^{K} \delta^{(k)}$,

1. $||\hat{\boldsymbol{\Pi}}_j - \boldsymbol{\Pi}_j||_2^2 \lesssim d \vee \log(1/\delta_{min}) \vee ||\boldsymbol{\pi}||_{2max}^2 / n_{min}$;

2. $||\mathbf{X}(\hat{\boldsymbol{\Pi}}_j - \boldsymbol{\Pi}_j)||_2^2 \lesssim d \vee \log(1/\delta_{min}) \vee ||\boldsymbol{\pi}||_{2max}^2$.

This implies that, with proper choice of the ridge parameter and sequence $\{\delta^{(k)}\}$, we can have well-bounded estimation and prediction loss, essentially $\ell_2$ consistent. For example, we can pick $\delta^{(k)} \asymp e^{-n_{\min}^t}$ where $t \in min(0.1 - \theta)$, so the $\ell_2$ estimation loss for each term and the MSE (Mean squared error) would tends to 0 with large sample size, as long as $||\boldsymbol{\pi}||_{2max}^2$ is bounded by a positive constant.

We could also draw conclusions in terms of the error over the whole system, i.e. the Frobenius norm. So that with probability at least $1 - p \sum_{k=1}^{K} \delta^{(k)}$, the systematic estimation error and MSE could reach the exact same bound for a single node. To control the loss with ultra-high probability, we only need control $p\delta^{(k)}$. Notice that each $p\delta^{(k)} \asymp pe^{-n_{\min}^t} \to 0$ whenever $p = o(e^{n_{min}^t})$. That is, the dimension can grow with a restricted exponential term, i.e, $p = e^{n_{min}^t}$.

After exploring the theory for the first stage, we will discuss the promising properties in the second stage. For each node $i$, we use $\mathcal{S}_i$ to denote the indices of true non-zero components of $\boldsymbol{\beta}_i$, i.e., $\mathcal{S}_i = \text{supp}(\boldsymbol{\beta}_i)$. Further denote

$$\boldsymbol{\Pi}_{-i} = \mathcal{T}(\boldsymbol{\pi}_{-i}^{(1)}, \boldsymbol{\pi}_{-i}^{(2)}, \cdots, \boldsymbol{\pi}_{-i}^{(K)}).$$

Following Bickel et al. [2], we impose the restricted eigenvalue condition to constrain the projected design matrix.

**Assumption B.3.** $||\mathbf{P}_i\mathbf{X}\boldsymbol{\Pi}_{-i}\beta_i||_2^2 \geq 2\lambda_0 n ||\beta_i||_2^2$ *whenever* $||\beta_{\mathcal{S}_i^c}||_1 \leq 3||\beta_{\mathcal{S}_i}||_1$ *for some positive constant* $\lambda_0$. *In addition,* $||\hat{\boldsymbol{\omega}}_{\mathcal{S}_i}||_\infty \leq ||\hat{\boldsymbol{\omega}}_{\mathcal{S}_i^c}||_{-\infty}$.

The latter assumption is intuitive. Recall $\hat{\boldsymbol{\omega}}_i = 1/|\hat{\boldsymbol{\beta}}_{0i}|^\gamma$, so we need $||\hat{\boldsymbol{\beta}}_{0\mathcal{S}_i}||_{-\infty} \geq ||\hat{\boldsymbol{\beta}}_{0\mathcal{S}_i^c}||_\infty$, where the estimation for true signal should always dominate the noise. This assumption is milder than the selection consistency of estimators which can be achieved by a proper estimator such as Lasso under mild conditions.

Denote

$$\mathbf{B} = [\boldsymbol{\beta}_1, \boldsymbol{\beta}_2, \ldots, \boldsymbol{\beta}_p],$$

and define

$$\begin{cases} f_n = \sqrt{n}||\boldsymbol{\Pi}||_1\sqrt{d \vee \log(\frac{1}{\delta_{min}}) \vee ||\boldsymbol{\pi}||_{2max}^2} + d \vee \log(\frac{1}{\delta_{min}}) \vee ||\boldsymbol{\pi}||_{2max}^2, \\ g_n = (C_\Pi\sqrt{\frac{d}{n_{min}}(d \vee \log(\frac{1}{\delta_{min}}) \vee ||\boldsymbol{\pi}||_{2max}^2)} + ||\boldsymbol{\Pi}||_1)(||\mathbf{B}||_1 \vee 1). \end{cases} \tag{9}$$

Next we will show that under some mild conditions, for each node $i$, the estimation and prediction loss could be bounded with large probability. Denote $\sigma_{max} = \max\limits_{1\leq i\leq p}\max\limits_{1\leq k\leq K}\sigma_i^{(k)}$ and $\tilde{\sigma}_{max} = \max\limits_{1\leq i\leq p}\max\limits_{1\leq k\leq K}\tilde{\sigma}_i^{(k)}$.

**Theorem B.3.** *Suppose that the adaptive lasso at the second stage takes the tuning parameter*

$$\nu_i = \frac{4}{\sqrt{n}||\hat{\boldsymbol{\omega}}_{\mathcal{S}_i}||_\infty}g_n max\{3\sqrt{2}\sqrt{\log(\frac{4Kd}{\delta_{min}})}(\sigma_{max} \vee \tilde{\sigma}_{max}), \sqrt{d \vee \log(\frac{1}{\delta_{min}}) \vee ||\boldsymbol{\pi}||_{2max}^2}\},$$

*and* $|\mathcal{S}_i| \leq \frac{\lambda_0}{C}\frac{n}{f_n}$ *for some constant $C$. Then under Assumptions B.1–B.3, we have that, with probability at least* $1 - \delta_{min} - p\sum_{k=1}^K \delta^{(k)}$,

1. $||\hat{\boldsymbol{\beta}}_i - \boldsymbol{\beta}_i||_2^2 \lesssim \frac{|S_i|}{n}g_n\{d \vee \log(\frac{1}{\delta_{min}}) \vee ||\boldsymbol{\pi}||_{2max}^2 \vee \log\frac{d}{\delta_{min}}\}$;

2. $\frac{1}{n}||\mathbf{P}_i\hat{\boldsymbol{\Upsilon}}_{-i}(\hat{\boldsymbol{\beta}}_i - \boldsymbol{\beta}_i)||_2^2 \lesssim \frac{|S_i|}{n}g_n\{d \vee \log(\frac{1}{\delta_{min}}) \vee ||\boldsymbol{\pi}||_{2max}^2 \vee \log\frac{d}{\delta_{min}}\}$.

For each $k$, we can separately conduct the analysis on each cohort and obtain the following property for single task estimation. That is, letting

$$f(n^{(k)}) = \sqrt{n^{(k)}}||\boldsymbol{\pi}^{(k)}||_1\sqrt{d \vee \log(\frac{1}{\delta^{(k)}}) \vee ||\boldsymbol{\pi}_i||_2^{2(k)}} + d \vee \log(\frac{1}{\delta^{(k)}}) \vee ||\boldsymbol{\pi}_i||_2^{2(k)},$$

$$g(n^{(k)}) = \left(C_\pi\sqrt{\frac{d}{n^{(k)}}(d \vee \log(\frac{1}{\delta^{(k)}}) \vee ||\boldsymbol{\pi}_i||_2^{2(k)})} + ||\boldsymbol{\pi}^{(k)}||_1\right)(||\mathbf{B}||_1 \vee 1),$$

we have the following result for each node $i$.

**Corollary B.3.1.** *Suppose that the adaptive lasso at the inference stage takes the tuning parameter*

$$\nu_i = \frac{4}{\sqrt{n^{(k)}}||\hat{\boldsymbol{\omega}}_{\mathcal{S}_i}||_\infty}g(n^{(k)})max\{3\sqrt{2}\sqrt{\log(\frac{4d}{\delta_{min}})}(\sigma_{max} \vee \tilde{\sigma}_{max}), \sqrt{d \vee \log(\frac{1}{\delta^{(k)}}) \vee ||\boldsymbol{\pi}_i||_2^{2(k)}}\},$$

*and* $|\mathcal{S}_i| \leq \frac{\lambda_0}{C}\frac{n^{(k)}}{f(n^{(k)})}$ *for some constant $C$. Then under Assumptions B.1–B.3, we have that, with probability at least* $1 - \delta_{min} - p\delta^{(k)}$,

1. $||\hat{\boldsymbol{\gamma}}_i^{(k)} - \boldsymbol{\gamma}_i^{(k)}||_2^2 \lesssim \frac{|S_i|}{n^{(k)}} g(n^{(k)})\{d \vee \log(\frac{1}{\delta^{(k)}}) \vee ||\boldsymbol{\pi}_i||_2^{2(k)} \vee \log \frac{d}{\delta_{min}}\}$;

2. $\frac{1}{n^{(k)}}||\mathbf{P}_i\hat{\boldsymbol{\Upsilon}}_{-i}(\hat{\boldsymbol{\gamma}}_i^{(k)} - \boldsymbol{\gamma}_i^{(k)})||_2^2 \lesssim \frac{|S_i|}{n^{(k)}} g(n^{(k)})\{d \vee \log(\frac{1}{\delta^{(k)}}) \vee ||\boldsymbol{\pi}_i||_2^{2(k)} \vee \log \frac{d}{\delta_{min}}\}$.

The proof directly follows by treating each cohort as a baseline cohort in theorem (4.1). Note that the denominator is making a crucial difference in inferring the bound. The algorithm indicates that when we aggregate the samples that does not differ too much, the estimator tends to converge faster, especially when $n \gg n^{(k)}$, where we have a plethora of samples in addition to the samples for the base cohort.

Moving forward, we will examine the selection consistency of the estimator. Denote the covariance matrix of $\mathbf{P}_i\mathbf{X}\boldsymbol{\Pi}_{-i}$ as

$$\Sigma_i = \frac{1}{n}\boldsymbol{\Pi}_{-i}^T\mathbf{X}^T\mathbf{P}_i\mathbf{X}\boldsymbol{\Pi}_{-i},$$

and correspondingly denote

$$\hat{\Sigma}_i = \frac{1}{n}\hat{\boldsymbol{\Pi}}_{-i}^T\mathbf{X}^T\mathbf{P}_i\mathbf{X}\hat{\boldsymbol{\Pi}}_{-i},$$

for $\mathbf{P}_i\mathbf{X}\hat{\boldsymbol{\Pi}}_{-i}$. We will use subscripts to denote the corresponding rows and columns in above matrices. For instance, $\Sigma_{S_i,S_i}$ represents the rows and columns of $\Sigma_i$, both indexed by $S_i$.

In order to investigate the selection consistency of causal effects, we impose the irrepresentable condition introduced by Zhao and Yu [24].

**Assumption B.4.** *For each $i \in \{1, 2, \cdots, p\}$, $\Sigma_{\mathcal{S}_i,\mathcal{S}_i}$ is invertible and $||\Sigma_{\mathcal{S}_i^c,\mathcal{S}_i}\Sigma_{\mathcal{S}_i,\mathcal{S}_i}^{-1}||_\infty < 1 - \eta$ for some constant $\eta \in (0, 1)$.*

**Theorem B.4.** *(Causality Selection Consistency) Suppose that $|\mathcal{S}_i| \leq \frac{\eta}{(\eta+2)\tau_i}\frac{n}{f_n}$ with $\tau_i = ||\Sigma_{\mathcal{S}_i,\mathcal{S}_i}^{-1}||_\infty$ and $\min_{j \in \mathcal{S}_i}|\boldsymbol{\beta}_{ij}| > \frac{\nu_i\tau_i||\hat{\boldsymbol{\omega}}_{\mathcal{S}_i}||_\infty}{2-\eta}$ for each node $i$. Then under Assumptions B.1–B.4, we have probability at least $1 - \delta_{min} - p\sum_{k=1}^K \delta^{(k)}$, such that $sign(\hat{\boldsymbol{\beta}}_i) = sign(\boldsymbol{\beta}_i)$.*

Note that estimation consistency does not imply sign consistency and vice versa. We further establish sign consistency to guarantee our causal variables are selected with the correct sign. From the discussion in Theorem 4.1, it directly follows that $\nu_i \lesssim n^{(1-3\theta)/2}$, which would infer $\frac{\nu_i\tau||\hat{\boldsymbol{\omega}}_{\mathcal{S}_i}||_\infty}{2-\eta} \lesssim n^{\frac{1-3\theta}{2}}$. Thus we are putting really mild assumption on the smallest true signal size, requiring at most a gap of $n^{\frac{2-3\theta}{2}}$ between the signal and the error decay rate of $n^{-\frac{1}{2}}$, As for the number of true significant variables, we can grant the growth at most to $n^{\frac{\theta}{2}}$.

## B.2 Proofs of the Theorems

### B.2.1 Proof of Theorem B.1

It can be inferred from [9] that there exists some $\theta^{(k)} \in (0, U^{(k)})$ such that, when $|\mathcal{M}_i^{(k)}| \lesssim (n^{(k)})^{1-\theta^{(k)}}$ and for some positive constant $\tilde{c}_4^{(k)}$, we have that, with probability at least $1 - \tilde{c}_4^{(k)}\exp\left\{-(n^{(k)})^{1-2\kappa^{(k)}}/\log(n^{(k)})\right\}$,

$$\mathcal{M}_{i*}^{(k)} \subseteq \mathcal{M}_i^{(k)}.$$

Denote

$$\theta = \min_{1 \leq k \leq K} \theta^{(k)},$$

then, for $|\mathcal{M}_i^{(k)}| \lesssim n_{\min}^{1-\theta}$, with probability at least $1 - \tilde{c}_4^{(k)}\exp\left\{-(n^{(k)})^{1-2\kappa_{max}}/\log(n^{(k)})\right\}$, we have that

$$\mathcal{M}_{i*}^{(k)} \subseteq \mathcal{M}_i^{(k)}$$

### B.2.2   Proof of Theorem B.2

In the proof of theorem 2, we will first give the bound of estimation loss and predictions loss for ridge regression on each network, after which we utilize the result to prove the bound for constructed networks.

**Lemma B.1.** *For each $k \in \{1, 2, \ldots, K\}$, set $\lambda_i^{(k)} = \sqrt{n^{(k)}}$. Under Assumptions B.1 and B.2, we have, with probability at least $1 - \delta^{(k)}$,*

1. $||\hat{\pi}_i^{(k)} - \pi_i^{(k)}||_2^2 \lesssim \frac{d \vee \log(1/\delta^{(k)}) \vee ||\pi_i^{(k)}||_2^2}{n^{(k)}}$;

2. $||\mathbf{X}_i^{(k)}(\hat{\pi}_i^{(k)} - \pi_i^{(k)})||_2^2 \lesssim d \vee \log(1/\delta^{(k)}) \vee ||\pi_i^{(k)}||_2^2.$

*Proof of Lemma B.1.* We first link the ridge regression estimator with the ordinary least squres (OLS) estimator denoted as

$$\hat{\pi}_i^{*(k)} = \big(\mathbf{X}_i^{(k)T}\mathbf{X}_i^{(k)}\big)^{-1}\mathbf{X}_i^{(k)T}\mathbf{Y}_i^{(k)}.$$

We write ridge estimator $\hat{\pi}_i^{(k)}$ as

$$\hat{\pi}_i^{(k)} = \big(\mathbf{X}_i^{(k)T}\mathbf{X}_i^{(k)} + \lambda_i^{(k)}I_d\big)^{-1}\mathbf{X}_i^{(k)T}\mathbf{Y}_i^{(k)} = L\hat{\pi}_i^{*(k)},$$

where

$$\begin{aligned}
L = L^T &= \big(\lambda_i^{(k)}(\mathbf{X}_i^{(k)T}\mathbf{X}_i^{(k)})^{-1} + I_d\big)^{-1} = \frac{1}{\lambda_i^{(k)}}\big((\mathbf{X}_i^{(k)T}\mathbf{X}_i^{(k)})^{-1} + \frac{1}{\lambda_i^{(k)}}I_d\big)^{-1} \\
&= I_d - \lambda_i^{(k)}(\mathbf{X}_i^{(k)T}\mathbf{X}_i^{(k)} + \lambda_i^{(k)}I_d)^{-1},
\end{aligned}$$

using Woodbury's identity.

The $\ell_2$ loss for estimation can be then decomposed as

$$\begin{aligned}
&||\hat{\pi}_i^{(k)} - \pi_i^{(k)}||_2^2 \\
&= (L\hat{\pi}_i^{*(k)} - \pi_i^{(k)})^T(L\hat{\pi}_i^{*(k)} - \pi_i^{(k)}) \\
&= \underbrace{(\hat{\pi}_i^{*(k)} - \pi_i^{(k)})^T L^T L(\hat{\pi}_i^{*(k)} - \pi_i^{(k)})}_{T_{21}} + \underbrace{2(\hat{\pi}_i^{*(k)} - \pi_i^{(k)})^T L^T(L - I_d)\pi_i^{(k)}}_{T_{22}} \\
&\quad + \underbrace{\pi_i^{(k)T}(L - I_d)^T(L - I_d)\pi_i^{(k)}}_{T_{23}}.
\end{aligned}$$

We will derive the bound for $T_{21}, T_{22}$ and $T_{23}$ with respectively, after which the estimation loss could be bound easily using the property of eigenvalues of $\mathbf{X}_i^{(k)T}\mathbf{X}_i^{(k)}$.

We write

$$\mathbf{X}_i^{(k)T}\mathbf{X}_i^{(k)} = Q_i^{(k)}V_i^{(k)}Q_i^{(k)T},$$

using eigendecomposition, where $Q_i^{(k)}$ is unitary, and $V_i^{(k)}$ is a diagonal matrix with diagonal entries to be eigenvalues $v_{ij}$, where $v_{ij} \asymp n^{(k)}$ according to Assumption B.2, for each $j \in \{1, 2, \ldots, d\}$. Then

$$L = \frac{1}{\lambda_i^{(k)}}Q_i^{(k)}\big(V_i^{(k)-1} + \frac{1}{\lambda_i^{(k)}}I_d\big)^{-1}Q_i^{(k)T} = I_d - \lambda_i^{(k)}Q_i^{(k)}\big(V_i^{(k)} + \lambda_i^{(k)}I_d\big)^{-1}Q_i^{(k)T}.$$

Denote

$$\mathrm{var}(\epsilon_{ji}^{(k)}) = \sigma_i^{(k)2}, \quad \mathrm{var}(\xi_{ji}^{(k)}) = \tilde{\sigma}_i^{(k)2}.$$

Note that

$$\hat{\pi}_i^{*(k)} - \pi_i^{(k)} \sim \mathcal{N}(0, \tilde{\sigma}_i^{(k)2}(\mathbf{X}_i^{(k)T}\mathbf{X}_i^{(k)})^{-1}),$$

which follows the sub-Gaussian distribution. Following Assumption B.2, the singular value of $\mathbf{I} - \mathbf{\Gamma}^{(k)}$ is bounded from below, then we have $\tilde{\sigma}_{max} \lesssim \sigma_{max}$. We then employ the Hanson-Wright inequality [19] to bound its tail. For any $t$, there is some positive constant $t_1$, such that

$$\mathbb{P}(T_{21} \geq \mathbb{E}(T_{21}) + t) \leq t_1 \exp\left(-\frac{t^2}{\tilde{\sigma}_i^{(k)4}/n^{(k)2}||L^TL||_F^2} \wedge \frac{t}{\tilde{\sigma}_i^{(k)2}/n^{(k)}||L^TL||_{op}}\right). \quad (10)$$

$$\mathbb{E}(T_{21}) = \tilde{\sigma}_i^{(k)2}\mathrm{tr}\left(LL^T\mathbf{X}_i^{(k)}\tilde{\mathbf{X}}_i^{(k)T}\right)$$

$$= \tilde{\sigma}_i^{(k)2}\mathrm{tr}\left(\frac{1}{\lambda_i^{(k)2}}Q_i^{(k)}(V_i^{(k)-1} + \frac{1}{\lambda_i^{(k)}}I_d)^{-2}V_i^{(k)-1}Q^{(k)T}\right)$$

$$= \tilde{\sigma}_i^{(k)2}\sum_{j=1}^d \frac{v_{ij}}{(v_{ij} + \lambda_i^{(k)})^2} \lesssim \frac{d}{n^{(k)}}. \quad (11)$$

$$||LL^T||_F^2 = \mathrm{tr}\left(LL^TLL^T\right)$$

$$= \mathrm{tr}\left(\frac{1}{\lambda_i^{(k)4}}Q_i^{(k)}(V_i^{(k)-1} + \frac{1}{\lambda_i^{(k)}}I_d)^{-4}Q^{(k)T}\right)$$

$$= \sum_{j=1}^d \frac{v_{ij}^4}{(v_{ij} + \lambda_i^{(k)})^4} \lesssim d. \quad (12)$$

$$||LL^T||_{op} = \sqrt{\lambda_{max}\left(LL^TLL^T\right)} \lesssim 1. \quad (13)$$

Let

$$t = \sqrt{\tilde{\sigma}_i^{(k)4}/n^{(k)2}||LL^T||_F^2 log(\frac{K}{\delta^{(k)}})/exp(t_1)} \vee \left(\tilde{\sigma}_i^{(k)2}/n^{(k)}||LL^T||_{op} log(\frac{K}{\delta^{(k)}})/exp(t_1)\right),$$

with (10), (11), (12), and (13), we have that, with probability at least $1 - \delta^{(k)}/K$,

$$T_{21} \lesssim \frac{d \vee \sqrt{d log(\frac{1}{\delta^{(k)}})} \vee log(\frac{1}{\delta^{(k)}})}{n^{(k)}}. \quad (14)$$

Next we will bound $T_{22}$ using Gaussian tail inequality. Denote

$$\boldsymbol{\pi}_i^{q(k)} = Q^{(k)T}\boldsymbol{\pi}_i^{(k)},$$

then

$$||\boldsymbol{\pi}_i^{q(k)}||_2^2 = ||\boldsymbol{\pi}_i^{(k)}||_2^2.$$

For any positive $t$,

$$\mathbb{P}\left(T_{22} \geq t\right) \leq \exp\left(-\frac{1}{2}\frac{t^2}{\mathrm{var}(T_{22})}\right).$$

where

$$\mathrm{var}(T_{22}) = 4\tilde{\sigma}_i^{(k)2}\boldsymbol{\pi}_i^{(k)T}(L - I_d)^T L(\mathbf{X}_i^{(k)T}\mathbf{X}_i^{(k)})^{-1}L^T(L - I_d)\boldsymbol{\pi}_i^{(k)}$$

$$= 4\tilde{\sigma}_i^{(k)2}\boldsymbol{\pi}_i^{(k)T}Q_i^{(k)}(V_i^{(k)} + \lambda_i^{(k)}I_d)^{-1}(V_i^{(k)-1} + \frac{1}{\lambda_i^{(k)}}I_d)^{-1}V_i^{(k)-1} \times$$

$$(V_i^{(k)-1} + \frac{1}{\lambda_i^{(k)}}I_d)^{-1}(V_i^{(k)} + \lambda_i^{(k)}I_d)^{-1}Q_i^{(k)T}\boldsymbol{\pi}_i^{(k)}$$

$$= 4\tilde{\sigma}_i^{(k)2}\sum_{j=1}^d \frac{\pi_i^{q(k)2}v_{ij}\lambda_i^{(k)2}}{(v_{ij} + \lambda_i^{(k)})^4} \lesssim \frac{||\boldsymbol{\pi}_i^{(k)}||_2^2}{n^{(k)2}}.$$

Let

$$t = \sqrt{2\mathrm{var}(T_{22})\log(K/\delta^{(k)})},$$

we have that, with probability at least $1 - \delta^{(k)}/K$,

$$T_{22} \lesssim \frac{||\boldsymbol{\pi}_i^{(k)}||_2}{n^{(k)}}\sqrt{\log(1/\delta^{(k)})}. \quad (15)$$

Finally, we can bound $T_{23}$ as

$$
\begin{aligned}
T_{23} &= \boldsymbol{\pi}_i^{(k)T} Q_i^{(k)} (V_i^{(k)} + \lambda_i^{(k)} I_d)^{-2} Q_i^{(k)T} \boldsymbol{\pi}_i^{(k)} \\
&= \lambda_i^{(k)2} \sum_{j=1}^d \frac{\pi_i^{q(k)2}}{(v_{ij} + \lambda_i^{(k)})^2} \lesssim \frac{||\boldsymbol{\pi}_i^{(k)}||_2^2}{n^{(k)}}.
\end{aligned}
\tag{16}
$$

Combining the bound (14), (15) and (16), we have that, with probability at least $1 - \delta^{(k)}$,

$$
||\hat{\boldsymbol{\pi}}_i^{(k)} - \boldsymbol{\pi}_i^{(k)}||_2^2 \lesssim \frac{d \vee log(\frac{1}{\delta^{(k)}}) \vee ||\boldsymbol{\pi}_i^{(k)}||_2^2}{n^{(k)}}.
$$

Note that

$$
\frac{(\hat{\boldsymbol{\pi}}_i^{(k)} - \boldsymbol{\pi}_i^{(k)})^T \mathbf{X}_i^{(k)T} \mathbf{X}_i^{(k)} (\hat{\boldsymbol{\pi}}_i^{(k)} - \boldsymbol{\pi}_i^{(k)})}{(\hat{\boldsymbol{\pi}}_i^{(k)} - \boldsymbol{\pi}_i^{(k)})^T (\hat{\boldsymbol{\pi}}_i^{(k)} - \boldsymbol{\pi}_i^{(k)})} \leq \lambda_{max}(\mathbf{X}_i^{(k)T} \mathbf{X}_i^{(k)}).
$$

So we directly have

$$
||\mathbf{X}_i^{(k)} (\hat{\boldsymbol{\pi}}_i^{(k)} - \boldsymbol{\pi}_i^{(k)})||_2^2 \leq \lambda_{max}(\mathbf{X}_i^{(k)T} \mathbf{X}_i^{(k)}) ||\hat{\boldsymbol{\pi}}_i^{(k)} - \boldsymbol{\pi}_i^{(k)}||_2^2 \lesssim d \vee log(\frac{1}{\delta^{(k)}}) \vee ||\boldsymbol{\pi}_i^{(k)}||_2^2.
$$

which concludes the proof of Lemma B.1. □

Next, we will prove the theorem by repeatedly applying Lemma B.1. We write the error loss for each $j \in \{1, 2, \ldots, Kp\}$ as

$$
||\hat{\boldsymbol{\Pi}}_j - \boldsymbol{\Pi}_j||_2^2 = \begin{cases} \sum_{k=1}^K ||\hat{\boldsymbol{\pi}}_{j|p}^{(k)} - \boldsymbol{\pi}_{j|p}^{(k)}||_2^2, & j \leq p, \\ ||\hat{\boldsymbol{\pi}}_{j|p}^{(k-1)} - \boldsymbol{\pi}_{j|p}^{(k-1)}||_2^2, & j > p, \end{cases}
\tag{17}
$$

where $j|d$ is the remainder of $j$ divided by $p$, denoting the corresponding column index. Applying the bounds in Lemma B.1 to all the networks, we have that, with probability at least $1 - \sum_{k=1}^K \delta^{(k)}$,

$$
||\hat{\boldsymbol{\Pi}}_j - \boldsymbol{\Pi}_j||_2^2 \lesssim \sum_{k=1}^K \frac{d \vee log(1/\delta^{(k)}) \vee ||\boldsymbol{\pi}_i^{(k)}||_2^2}{n^{(k)}} \lesssim \frac{d \vee log(1/\delta_{min}) \vee ||\boldsymbol{\pi}||_{2max}^2}{n_{min}}.
$$

Using the same approach we can bound the prediction loss. We have that, with probability at least $1 - \sum_{k=1}^K \delta^{(k)}$,

$$
\begin{aligned}
||\mathbf{X}(\hat{\boldsymbol{\Pi}}_j - \boldsymbol{\Pi}_j)||_2^2 &= \sum_{k=1}^K ||X_i^{(k)} (\hat{\boldsymbol{\pi}}_{j|p}^{(k)} - \boldsymbol{\pi}_{j|p}^{(k)})||_2^2 \\
&\lesssim \sum_{k=1}^K d \vee log(1/\delta^{(k)}) \vee ||\boldsymbol{\pi}_i^{(k)}||_2^2 \lesssim d \vee log(1/\delta_{min}) \vee ||\boldsymbol{\pi}||_{2max}^2,
\end{aligned}
$$

which concludes the proof of Theorem B.2.

### B.2.3  Proof of Theorem B.3

**Lemma B.2.** *Suppose that for each node $i$ and a properly chosen positive constant $C$,*

$$
|\mathcal{S}_i| \leq \frac{\lambda_0}{C} \frac{n}{f_n}.
\tag{18}
$$

*for function $f_n$ defined in (9). Under Assumptions B.1–B.3, we have that, with probability at least $1 - p \sum_{k=1}^K \delta^{(k)}$,*

$$
||\mathbf{P}_i \mathbf{X} \hat{\boldsymbol{\Pi}}_{-i} \beta_i||_2^2 \geq \lambda_0 n ||\beta_i||_2^2,
$$

*whenever $||\beta_{S_i^c}||_1 \leq 3||\beta_{S_i}||_1$.*

*Proof of Lemma B.2.* We will bound the maximum value in the matrix

$$|(\mathbf{P}_i\mathbf{X}\hat{\mathbf{\Pi}}_{-i})^T(\mathbf{P}_i\mathbf{X}\hat{\mathbf{\Pi}}_{-i}) - (\mathbf{P}_i\mathbf{X}\mathbf{\Pi}_{-i})^T(\mathbf{P}_i\mathbf{X}\mathbf{\Pi}_{-i})|,$$

whose value in $l$-th row and $r$-th column can be expressed as

$$
\begin{aligned}
&|\hat{\mathbf{\Pi}}_l^T\mathbf{X}\mathbf{P}_i\mathbf{X}\hat{\mathbf{\Pi}}_r - \mathbf{\Pi}_l^T\mathbf{X}\mathbf{P}_i\mathbf{X}\mathbf{\Pi}_r| \\
&\leq \ |\underbrace{(\hat{\mathbf{\Pi}}_l - \mathbf{\Pi}_l)^T\mathbf{X}^T\mathbf{P}_i\mathbf{X}(\hat{\mathbf{\Pi}}_r - \mathbf{\Pi}_r)}_{T_{31}}| + |\underbrace{(\hat{\mathbf{\Pi}}_l - \mathbf{\Pi}_l)^T\mathbf{X}^T\mathbf{P}_i\mathbf{X}\mathbf{\Pi}_r}_{T_{32}}| \\
&+ \ |\underbrace{\mathbf{\Pi}_l^T\mathbf{X}^T\mathbf{P}_i\mathbf{X}(\hat{\mathbf{\Pi}}_r - \mathbf{\Pi}_r)}_{T_{33}}|.
\end{aligned}
$$

We will further bound $T_{31}, T_{32}$ and $T_{33}$. Note that, since $\mathbf{P}_i$ is a projection matrix,

$$\lambda_{max}(\mathbf{P}_i) = 1.$$

Then by applying Theorem B.2, we have that, with probability at least $1 - \sum_{k=1}^{K}\delta^{(k)}$,

$$
\begin{aligned}
T_{31} &\leq \ ||\mathbf{P}_i\mathbf{X}(\hat{\mathbf{\Pi}}_l - \mathbf{\Pi}_l)||_2 \times ||\mathbf{P}_i\mathbf{X}(\hat{\mathbf{\Pi}}_r - \mathbf{\Pi}_r)||_2 \\
&\leq \ \lambda_{max}^2(\mathbf{P}_i)||\mathbf{X}(\hat{\mathbf{\Pi}}_l - \mathbf{\Pi}_l)||_2 \times ||\mathbf{X}(\hat{\mathbf{\Pi}}_r - \mathbf{\Pi}_r)||_2 \\
&\lesssim \ d \vee \log(\frac{1}{\delta_{min}}) \vee ||\boldsymbol{\pi}||_{2max}^2.
\end{aligned}
$$

With equation (17), we have that,

$$||\mathbf{X}\,\mathbf{\Pi}_r||_2^2 \leq \sum_{k=1}^{K}||X_i^{(k)}\boldsymbol{\pi}_{r|p}^{(k)}||_2^2 \lesssim \sum_{i=1}^{K}n^{(k)}||\boldsymbol{\pi}_{r|p}^{(k)}||_2^2 \lesssim n\sum_{i=1}^{K}||\boldsymbol{\pi}_{r|p}^{(k)}||_2^2 \lesssim n||\mathbf{\Pi}||_1^2.$$

Thus $T_{32}$ could be bounded as

$$T_{32} \leq ||\mathbf{X}\mathbf{\Pi}_r||_2||\mathbf{P}_i\mathbf{X}(\hat{\mathbf{\Pi}}_l - \mathbf{\Pi}_l)||_2 \lesssim \sqrt{n}||\mathbf{\Pi}||_1\sqrt{d \vee \log(\frac{1}{\delta_{min}}) \vee ||\boldsymbol{\pi}||_{2max}^2}. \tag{19}$$

In a similar manner, we have

$$T_{33} \lesssim \sqrt{n}||\mathbf{\Pi}||_1\sqrt{d \vee \log(\frac{1}{\delta_{min}}) \vee ||\boldsymbol{\pi}||_{2max}^2}. \tag{20}$$

Then

$$T_{31} + T_{32} + T_{33} \lesssim \sqrt{n}||\mathbf{\Pi}||_1\sqrt{d \vee \log(\frac{1}{\delta_{min}}) \vee ||\boldsymbol{\pi}||_{2max}^2} + d \vee \log(\frac{1}{\delta_{min}}) \vee ||\boldsymbol{\pi}||_{2max}^2 = f_n.$$

Let, for certain positive constant $C$,

$$T_{31} + T_{32} + T_{33} \leq Cf_n,$$

then we have,

$$
\begin{aligned}
&\beta_i^T((\mathbf{P}_i\mathbf{X}\hat{\mathbf{\Pi}}_{-i})^T(\mathbf{P}_i\mathbf{X}\hat{\mathbf{\Pi}}_{-i}) - (\mathbf{P}_i\mathbf{X}\mathbf{\Pi}_{-i})^T(\mathbf{P}_i\mathbf{X}\mathbf{\Pi}_{-i}))\beta_i \\
&\leq \ ||\beta_i||_1^2(|T_{31}| + |T_{32}| + |T_{33}|) \leq C|\mathcal{S}_i|||\beta_i||_2^2 f_n \leq \lambda_0 n||\beta_i||_2^2. \tag{21}
\end{aligned}
$$

Along with the restriction of Assumption B.3, we have

$$||\mathbf{P}_i\mathbf{X}\hat{\mathbf{\Pi}}_{-i}\beta_i||_2^2 \geq \lambda_0 n||\beta_i||_2^2,$$

whenever $||\beta_{S_i^c}||_1 \leq 3||\beta_{S_i}||_1$ by applying triangular inequality. The proof of Lemma B.2 is now complete. □

**Lemma B.3.** *For each $i \in \{1, 2, \ldots, p\}$, $\nu_i$ is the tuning parameter in the inference stage. Under Assumptions B.1–B.3, we have, with probability at least $1 - \delta_{min} - p\sum_{k=1}^{K}\delta^{(k)}$,*

$$||\hat{\mathbf{\Upsilon}}_{-i}^T\mathbf{P}_i[\boldsymbol{\epsilon}_i + (\mathbf{\Upsilon}_{-i} - \hat{\mathbf{\Upsilon}}_{-i})\boldsymbol{\beta_i}]||_\infty \leq \frac{1}{4}n\nu_i||\hat{\boldsymbol{\omega}}_{S_i}||_\infty. \tag{22}$$

*Proof of Lemma B.3.* We have $\boldsymbol{\Upsilon}_{-i} = \mathbf{X}\boldsymbol{\Pi}_{-i} + \boldsymbol{\xi}_{-i}$ and $\hat{\boldsymbol{\Upsilon}}_{-i} = \mathbf{X}\hat{\boldsymbol{\Pi}}_{-i}$ by definition, where
$$\boldsymbol{\xi}_{-i} = \mathcal{T}(\boldsymbol{\xi}_{-i}^{(1)}, \boldsymbol{\xi}_{-i}^{(2)}, \cdots, \boldsymbol{\xi}_{-i}^{(K)}).$$

Note that
$$||\hat{\boldsymbol{\Upsilon}}_{-i}^T \mathbf{P}_i[\boldsymbol{\epsilon}_i + (\boldsymbol{\Upsilon}_{-i} - \hat{\boldsymbol{\Upsilon}}_{-i})\boldsymbol{\beta}_i]||_\infty \leq \underbrace{||\hat{\boldsymbol{\Pi}}_{-i}^T \mathbf{X}^T \mathbf{P}_i \boldsymbol{\epsilon}_i||_\infty}_{T_{34}} + \underbrace{||\hat{\boldsymbol{\Pi}}_{-i}^T \mathbf{X}^T \mathbf{P}_i \mathbf{X}(\boldsymbol{\Pi}_{-i}^T - \hat{\boldsymbol{\Pi}}_{-i}^T)\boldsymbol{\beta}_i||_\infty}_{T_{35}}$$
$$+ \underbrace{||\hat{\boldsymbol{\Pi}}_{-i}^T \mathbf{X}^T \mathbf{P}_i \boldsymbol{\xi}_{-i} \boldsymbol{\epsilon}_i||_\infty}_{T_{36}}.$$

Firstly, by applying Theorem B.2, we have that, with probability at least $1 - p\sum_{k=1}^K \delta^{(k)}$,
$$||(\hat{\boldsymbol{\Pi}}_j - \boldsymbol{\Pi}_j)^T||_{-\infty}^2 = \max_{1 \leq j \leq Kp} ||\hat{\boldsymbol{\Pi}}_j - \boldsymbol{\Pi}_j||_1^2 \leq \max_{1 \leq j \leq Kp} \left( Kd||\hat{\boldsymbol{\Pi}}_j - \boldsymbol{\Pi}_j||_2^2 \right)$$
$$\lesssim \frac{d}{n_{min}}(d \vee \log(\frac{1}{\delta_{min}}) \vee ||\boldsymbol{\pi}||_{2max}^2).$$

Denote $t_{\Pi} = ||(\hat{\boldsymbol{\Pi}}_j - \boldsymbol{\Pi}_j)^T||_{-\infty} \leq C_\Pi \sqrt{\frac{d}{n_{min}}(d \vee \log(\frac{1}{\delta_{min}}) \vee ||\boldsymbol{\pi}||_{2max}^2)}$,

We further have $X_{.j}^T X_{.j} \asymp n$ after standardization, where $X_{.j}$ is the $j$-th column of $\mathbf{X}$. Therefore, $X_{.j}^T \mathbf{P}_i \boldsymbol{\epsilon}_i$ follows Gaussian distribution with variance bounded as
$$\text{var}\left(X_{.j}^T \mathbf{P}_i \boldsymbol{\epsilon}_i\right) \lesssim X_{.j}^T \mathbf{P}_i X_{.j} \lesssim n.$$

We have
$$\mathbb{P}\left(||\hat{\boldsymbol{\Pi}}_{-i}^T \mathbf{X}^T \mathbf{P}_i \boldsymbol{\epsilon}_i||_\infty \geq t\right)$$
$$\leq \mathbb{P}\left((||(\hat{\boldsymbol{\Pi}}_{-i} - \boldsymbol{\Pi}_{-i})^T||_\infty + ||\boldsymbol{\Pi}_{-i}^T||_\infty)||\mathbf{X}^T \mathbf{P}_i \boldsymbol{\epsilon}_i||_\infty \geq t\right)$$
$$\leq \mathbb{P}\left(||\mathbf{X}^T \mathbf{P}_i \boldsymbol{\epsilon}_i||_\infty \geq \frac{t}{t_\Pi + ||\boldsymbol{\Pi}||_1}\right)$$
$$\leq 2Kd \exp\left\{-\left(\frac{t}{t_\Pi + ||\boldsymbol{\Pi}||_1}\right)^2 / (2n\sigma_{max}^2)\right\}. \tag{23}$$

Using Theorem B.2 and Assumption B.2, the second term could be bounded as,
$$||\hat{\boldsymbol{\Pi}}_{-i}^T \mathbf{X}^T \mathbf{P}_i \mathbf{X}(\boldsymbol{\Pi}_{-i}^T - \hat{\boldsymbol{\Pi}}_{-i}^T)\boldsymbol{\beta}_i||_\infty$$
$$\leq (||(\hat{\boldsymbol{\Pi}}_{-i} - \boldsymbol{\Pi}_{-i})^T||_\infty + ||\boldsymbol{\Pi}_{-i}^T||_\infty)||\mathbf{X}^T \mathbf{P}_i \mathbf{X}(\boldsymbol{\Pi}_{-i}^T - \hat{\boldsymbol{\Pi}}_{-i}^T)\boldsymbol{\beta}_i||_\infty$$
$$\leq (t_\Pi + ||\boldsymbol{\Pi}||_1)||\mathbf{B}||_1 \max_{j_1,j_2} |\mathbf{X}_{j_1}^T \mathbf{P}_i \mathbf{X}(\boldsymbol{\Pi}_{j_2}^T - \hat{\boldsymbol{\Pi}}_{j_2}^T)|$$
$$\leq \sqrt{n}(t_\Pi + ||\boldsymbol{\Pi}||_1)||\mathbf{B}||_1 \max_{j_2} ||\mathbf{P}_i \mathbf{X}(\boldsymbol{\Pi}_{j_2}^T - \hat{\boldsymbol{\Pi}}_{j_2}^T)||_2$$
$$\leq \sqrt{n}(t_\Pi + ||\boldsymbol{\Pi}||_1)||\mathbf{B}||_1 \sqrt{d \vee \log(\frac{1}{\delta_{min}}) \vee ||\boldsymbol{\pi}||_{2max}^2}, \tag{24}$$

Using the same way above, we have
$$\text{var}(X_{.j}^T \mathbf{P}_i \boldsymbol{\xi}_i) \lesssim n.$$
Then,
$$\mathbb{P}\left(||\hat{\boldsymbol{\Pi}}_{-i}^T \mathbf{X}^T \mathbf{P}_i \boldsymbol{\xi}_{-i} \boldsymbol{\epsilon}_i \boldsymbol{\beta}_i||_\infty \geq t\right)$$
$$\leq \mathbb{P}\left((||(\hat{\boldsymbol{\Pi}}_{-i} - \boldsymbol{\Pi}_{-i})^T||_\infty + ||\boldsymbol{\Pi}_{-i}^T||_\infty)||\mathbf{B}||_1 ||\mathbf{X}^T \mathbf{P}_i \boldsymbol{\xi}_{-i} \boldsymbol{\epsilon}_i||_\infty \geq t\right)$$
$$\leq \mathbb{P}\left(||\mathbf{X}^T \mathbf{P}_i \boldsymbol{\xi}_{-i} \boldsymbol{\epsilon}_i||_\infty \geq \frac{t}{(t_\Pi + ||\boldsymbol{\Pi}||_1)||\mathbf{B}||_1}\right)$$
$$\leq 2Kd \exp\left\{-\left(\frac{t}{(t_\Pi + ||\boldsymbol{\Pi}||_1)||\mathbf{B}||_1}\right)^2 / (2n\tilde{\sigma}_{max}^2)\right\}. \tag{25}$$

With $\nu_i$ defined in Theorem B.3 and

$$t = \frac{1}{12}n\nu_i\|\hat{\boldsymbol{\omega}}_{S_i}\|_\infty$$

in the above inequalities, we then have

$$\mathbb{P}(T_{34} \geq \frac{1}{12}n\nu_i\|\hat{\boldsymbol{\omega}}_{S_i}\|_\infty) \leq \delta_{min}/2,$$

$$\mathbb{P}(T_{36} \geq \frac{1}{12}n\nu_i\|\hat{\boldsymbol{\omega}}_{S_i}\|_\infty) \leq \delta_{min}/2,$$

$$T_{35} \leq \frac{1}{12}n\nu_i\|\hat{\boldsymbol{\omega}}_{S_i}\|_\infty.$$

Conditioning on $t_\Pi$ and using the bound obtained at the beginning of the discussion. We have that

$$\mathbb{P}(T_{34} + T_{35} + T_{36} \leq \frac{1}{4}n\nu_i\|\hat{\boldsymbol{\omega}}_{S_i}\|_\infty) \geq (1 - \delta_{min} - p\sum_{k=1}^{K}\delta^{(k)}).$$

This concludes the proof of Lemma B.3. $\qquad\qquad\qquad\qquad\qquad\square$

Next, we follow the techniques in [17] to bound the prediction loss. Following the definition of the adaptive lasso, we have

$$\frac{1}{n}\|\mathbf{P}_i\mathbf{Y}_i - \mathbf{P}_i\hat{\boldsymbol{\Upsilon}}_{-i}\hat{\boldsymbol{\beta}}_i\|_2^2 + \nu_i\hat{\boldsymbol{\omega}}_i^T|\hat{\boldsymbol{\beta}}_i|_1 \leq \frac{1}{n}\|\mathbf{P}_i\mathbf{Y}_i - \mathbf{P}_i\hat{\boldsymbol{\Upsilon}}_{-i}\boldsymbol{\beta}_i\|_2^2 + \nu_i\hat{\boldsymbol{\omega}}_i^T|\boldsymbol{\beta}_i|_1. \qquad (26)$$

We can rewrite the inequality as

$$\frac{1}{n}\|\mathbf{P}_i(\boldsymbol{\Upsilon}_{-i} - \hat{\boldsymbol{\Upsilon}}_{-i})\boldsymbol{\beta}_i + \mathbf{P}_i\boldsymbol{\epsilon}_i + \mathbf{P}\hat{\boldsymbol{\Upsilon}}_{-i}(\boldsymbol{\beta}_i - \hat{\boldsymbol{\beta}}_i)\|_2^2 + \nu_i\hat{\boldsymbol{\omega}}_i^T|\hat{\boldsymbol{\beta}}_i|_1$$

$$\leq \frac{1}{n}\|\mathbf{P}_i(\boldsymbol{\Upsilon}_{-i} - \hat{\boldsymbol{\Upsilon}}_{-i})\boldsymbol{\beta}_i + \mathbf{P}_i\boldsymbol{\epsilon}_i\|_2^2 + \nu_i\hat{\boldsymbol{\omega}}_i^T|\boldsymbol{\beta}_i|_1,$$

$$\frac{1}{n}\|\mathbf{P}_i(\boldsymbol{\Upsilon}_{-i} - \hat{\boldsymbol{\Upsilon}}_{-i})\boldsymbol{\beta}_i\|_2^2$$

$$\leq \frac{2}{n}\{\hat{\boldsymbol{\Upsilon}}_{-i}^T\mathbf{P}_i[\boldsymbol{\epsilon}_i + (\boldsymbol{\Upsilon}_{-i} - \hat{\boldsymbol{\Upsilon}}_{-i})\boldsymbol{\beta_i}]\}^T(\hat{\boldsymbol{\beta}}_i - \boldsymbol{\beta}_i) + \nu_i\hat{\boldsymbol{\omega}}_i^T|\boldsymbol{\beta}_i|_1 - \nu_i\hat{\boldsymbol{\omega}}_i^T|\hat{\boldsymbol{\beta}}_i|_1.$$

Adding $\frac{1}{2}\nu_i\|\hat{\boldsymbol{\omega}}_{S_i}\|_\infty\|\hat{\boldsymbol{\beta}}_i - \boldsymbol{\beta}_i\|_1$, and then multiplying $n$ to both sides, together with Lemma B.3 and Assumption B.3, we have that

$$\|\mathbf{P}\hat{\boldsymbol{\Upsilon}}_{-i}(\boldsymbol{\beta}_i - \hat{\boldsymbol{\beta}}_i)\|_2^2 + \frac{n}{2}\nu_i\|\hat{\boldsymbol{\omega}}_{S_i}\|_\infty\|\hat{\boldsymbol{\beta}}_i - \boldsymbol{\beta}_i\|_1$$

$$\leq \quad 2\|\{\hat{\boldsymbol{\Upsilon}}_{-i}^T\mathbf{P}_i[\boldsymbol{\epsilon}_i + (\boldsymbol{\Upsilon}_{-i} - \hat{\boldsymbol{\Upsilon}}_{-i})\boldsymbol{\beta_i}]\}^T\|_\infty\|(\hat{\boldsymbol{\beta}}_i - \boldsymbol{\beta}_i)\|_1 + \frac{n}{2}\nu_i\|\hat{\boldsymbol{\omega}}_{S_i}\|_\infty\|\hat{\boldsymbol{\beta}}_i - \boldsymbol{\beta}_i\|_1$$

$$\quad + n\nu_i\hat{\boldsymbol{\omega}}_i^T|\boldsymbol{\beta}_i|_1 - n\nu_i\hat{\boldsymbol{\omega}}_i^T|\hat{\boldsymbol{\beta}}_i|_1$$

$$\leq \quad n\nu_i\|\hat{\boldsymbol{\omega}}_{S_i}\|_\infty\|\hat{\boldsymbol{\beta}}_i - \boldsymbol{\beta}_i\|_1 + n\nu_i\|\hat{\boldsymbol{\omega}}_{S_i}\|_\infty\|\boldsymbol{\beta}_{S_i}\|_1 - n\nu_i\|\hat{\boldsymbol{\omega}}_{S_i}\|_\infty(\|\hat{\boldsymbol{\beta}}_{S_i}\|_1 + \|\hat{\boldsymbol{\beta}}_{S_i^C}\|_1)$$

$$= \quad n\nu_i\|\hat{\boldsymbol{\omega}}_{S_i}\|_\infty\|\hat{\boldsymbol{\beta}}_{S_i} - \boldsymbol{\beta}_{S_i}\|_1 + n\nu_i\|\hat{\boldsymbol{\omega}}_{S_i}\|_\infty\|\boldsymbol{\beta}_{S_i}\|_1 - n\nu_i\|\hat{\boldsymbol{\omega}}_{S_i}\|_\infty\|\hat{\boldsymbol{\beta}}_{S_i}\|_1$$

$$\leq \quad 2n\nu_i\|\hat{\boldsymbol{\omega}}_{S_i}\|_\infty\|\hat{\boldsymbol{\beta}}_{S_i} - \boldsymbol{\beta}_{S_i}\|_1. \qquad (27)$$

Comparing the two sides, we have

$$\|\hat{\boldsymbol{\beta}}_i - \boldsymbol{\beta}_i\|_1 \leq 4\|\hat{\boldsymbol{\beta}}_{\mathcal{S}_i} - \boldsymbol{\beta}_{\mathcal{S}_i}\|_1, \qquad (28)$$

$$\|\hat{\boldsymbol{\beta}}_{\mathcal{S}_i^c} - \boldsymbol{\beta}_{\mathcal{S}_i^c}\|_1 \leq 3\|\hat{\boldsymbol{\beta}}_{\mathcal{S}_i} - \boldsymbol{\beta}_{\mathcal{S}_i}\|_1. \qquad (29)$$

which indicates that $\hat{\boldsymbol{\beta}} - \boldsymbol{\beta}$ satisfies the condition in Lemma B.2. So we have

$$\|\mathbf{P}_i\hat{\boldsymbol{\Upsilon}}_{-i}(\hat{\boldsymbol{\beta}}_i - \boldsymbol{\beta}_i)\|_2^2$$

$$\leq \quad \frac{3}{2}\nu_in\|\hat{\boldsymbol{\omega}}_{\mathcal{S}_i}\|_\infty\|\hat{\boldsymbol{\beta}}_{\mathcal{S}_i} - \boldsymbol{\beta}_{\mathcal{S}_i}\|_1 \leq \frac{3}{2}\nu_in\|\hat{\boldsymbol{\omega}}_{\mathcal{S}_i}\|_\infty\sqrt{|\mathcal{S}_i|}\|\hat{\boldsymbol{\beta}}_{\mathcal{S}_i} - \boldsymbol{\beta}_{\mathcal{S}_i}\|_2$$

$$\leq \quad \frac{3}{2}\nu_in\|\hat{\boldsymbol{\omega}}_{\mathcal{S}_i}\|_\infty\sqrt{|\mathcal{S}_i|}\frac{\|\mathbf{P}_i\hat{\boldsymbol{\Upsilon}}_{-i}(\hat{\boldsymbol{\beta}}_i - \boldsymbol{\beta}_i)\|_2}{\sqrt{n\lambda_0}}. \qquad (30)$$

Thus we can bound the error term as,

$$\frac{1}{n}||\mathbf{P}_i\hat{\mathbf{\Upsilon}}_{-i}(\hat{\boldsymbol{\beta}}_i - \boldsymbol{\beta}_i)||_2^2 \leq \frac{9}{4}\frac{|\mathcal{S}_i|}{\lambda_0}||\hat{\boldsymbol{\omega}}_{\mathcal{S}_i}||_\infty^2\nu_i^2.$$

Using the value of $\nu_i$, we get that

$$\frac{1}{n}||\mathbf{P}_i\hat{\mathbf{\Upsilon}}_{-i}(\hat{\boldsymbol{\beta}}_i - \boldsymbol{\beta}_i)||_2^2$$

$$\lesssim \quad \frac{|S_i|}{n}(\sqrt{d\frac{d\vee\log(\frac{1}{\delta_{min}})\vee||\boldsymbol{\pi}||_{2max}^2}{n_{min}}}$$

$$+||\boldsymbol{\Pi}||_1)(||\mathbf{B}||_1\vee 1)\times d\vee\log(\frac{1}{\delta_{min}})\vee||\boldsymbol{\pi}||_{2max}^2\vee log\frac{d}{\delta_{min}}.$$

Applying Lemma B.2 again, we derive that

$$||\hat{\boldsymbol{\beta}}_i - \boldsymbol{\beta}_i||_2^2$$

$$\leq \quad \frac{1}{\lambda_0 n}||\mathbf{P}_i\hat{\mathbf{\Upsilon}}_{-i}(\hat{\boldsymbol{\beta}}_i - \boldsymbol{\beta}_i)||_2^2$$

$$\lesssim \quad \frac{|S_i|}{n}(\sqrt{d\frac{d\vee\log(\frac{1}{\delta_{min}})\vee||\boldsymbol{\pi}||_{2max}^2}{n_{min}}}$$

$$+||\boldsymbol{\Pi}||_1)\times(||\mathbf{B}||_1\vee 1)d\vee\log(\frac{1}{\delta_{min}})\vee||\boldsymbol{\pi}||_{2max}^2\vee log\frac{d}{\delta_{min}}.$$

The above prediction and estimation bounds condition on the bound of $t_\Pi$ and restricted eigenvalue condition for prediction matrices, which hold with probability at least $1 - \delta_{min} - p\sum_{k=1}^K \delta^k$. The proof of Theorem B.3 is then completed.

### B.2.4 Proof of Theorem B.4

**Lemma B.4.** *Suppose that, for each node $i$ and function $f_n$ defined in (9),*

$$|\mathcal{S}_i| \leq \frac{\eta}{(\eta+2)\tau_i}\frac{n}{f_n}. \tag{31}$$

*Under Assumptions B.1–B.4, we have that, with the probability at least $1 - p\sum_{k=1}^K$,*

$$||\hat{\Sigma}_{\mathcal{S}_i^c,\mathcal{S}_i}\hat{\Sigma}_{\mathcal{S}_i,\mathcal{S}_i}^{-1}||_\infty \leq 1 - \eta^2/2.$$

*Proof of Lemma B.4.* Following the proof of Lemma B.2, we have showed that, with probability at least $1 - p\sum_{k=1}^K \delta^{(k)}$,

$$\max_{l,r}\frac{1}{n}|\hat{\Sigma}_{l,r} - \Sigma_{l,r}| \leq f_n/n.$$

where the subscript $l, r$ denotes the elements in the $l$-th row and $r$-th column in the corresponding matrix.

Consider the part indexed by set $\mathcal{S}_i$ with the assumption in the lemma, we have that

$$||\hat{\Sigma}_{\mathcal{S}_i,\mathcal{S}_i} - \Sigma_{\mathcal{S}_i,\mathcal{S}_i}||_\infty \leq |\mathcal{S}_i|\frac{f_n}{n} \leq \frac{\eta}{(\eta+2)\tau_i}.$$

In a similar manner, we have,

$$||\hat{\Sigma}_{\mathcal{S}_i^c,\mathcal{S}_i} - \Sigma_{\mathcal{S}_i^c,\mathcal{S}_i}||_\infty \leq |\mathcal{S}_i|\frac{f_n}{n} \leq \frac{\eta}{(\eta+2)\tau_i}. \tag{32}$$

We bound the error of matrix inversion as described in Horn and Johnson [12] and obtain that,

$$||\hat{\Sigma}_{\mathcal{S}_i,\mathcal{S}_i}^{-1}||_\infty \leq ||\hat{\Sigma}_{\mathcal{S}_i,\mathcal{S}_i}^{-1} - \Sigma_{\mathcal{S}_i,\mathcal{S}_i}^{-1}||_\infty + ||\Sigma_{\mathcal{S}_i,\mathcal{S}_i}^{-1}||_\infty$$

$$\leq \frac{\tau_i^2||\hat{\Sigma}_{\mathcal{S}_i,\mathcal{S}_i} - \Sigma_{\mathcal{S}_i,\mathcal{S}_i}||_\infty}{1 - \tau_i||\hat{\Sigma}_{\mathcal{S}_i,\mathcal{S}_i} - \Sigma_{\mathcal{S}_i,\mathcal{S}_i}||_\infty} + \tau_i$$

$$\leq \frac{\tau_i}{1 - \tau_i|\mathcal{S}_i|f_n/n} \leq \frac{\eta+2}{2}\tau_i. \tag{33}$$

Note the following decomposition,

$$
\begin{aligned}
\hat{\Sigma}_{\mathcal{S}_i^c,\mathcal{S}_i}\hat{\Sigma}_{\mathcal{S}_i,\mathcal{S}_i}^{-1} - \Sigma_{\mathcal{S}_i^c,\mathcal{S}_i}\Sigma_{\mathcal{S}_i,\mathcal{S}_i}^{-1} &= (\hat{\Sigma}_{\mathcal{S}_i^c,\mathcal{S}_i} - \Sigma_{\mathcal{S}_i^c,\mathcal{S}_i})\hat{\Sigma}_{\mathcal{S}_i,\mathcal{S}_i}^{-1} \\
&+ \Sigma_{\mathcal{S}_i^c,\mathcal{S}_i}\Sigma_{\mathcal{S}_i,\mathcal{S}_i}^{-1}\left(\Sigma_{\mathcal{S}_i,\mathcal{S}_i} - \hat{\Sigma}_{\mathcal{S}_i,\mathcal{S}_i}\right)\hat{\Sigma}_{\mathcal{S}_i,\mathcal{S}_i}^{-1}.
\end{aligned}
$$

Then, collecting (32), (32), (33) and Assumption B.4, we have that

$$
\begin{aligned}
||\hat{\Sigma}_{\mathcal{S}_i^c,\mathcal{S}_i}\hat{\Sigma}_{\mathcal{S}_i,\mathcal{S}_i}^{-1} - \Sigma_{\mathcal{S}_i^c,\mathcal{S}_i}\Sigma_{\mathcal{S}_i,\mathcal{S}_i}^{-1}||_\infty &\leq ||\hat{\Sigma}_{\mathcal{S}_i^c,\mathcal{S}_i} - \Sigma_{\mathcal{S}_i^c,\mathcal{S}_i}||_\infty||\hat{\Sigma}_{\mathcal{S}_i,\mathcal{S}_i}^{-1}||_\infty \\
&+ ||\Sigma_{\mathcal{S}_i^c,\mathcal{S}_i}\Sigma_{\mathcal{S}_i,\mathcal{S}_i}^{-1}||_\infty||\hat{\Sigma}_{\mathcal{S}_i,\mathcal{S}_i} - \Sigma_{\mathcal{S}_i,\mathcal{S}_i}||_\infty||\hat{\Sigma}_{\mathcal{S}_i,\mathcal{S}_i}^{-1}||_\infty \\
&\leq \eta - \frac{1}{2}\eta^2.
\end{aligned}
$$

Together with Assumption B.4, we derive that

$$
||\hat{\Sigma}_{\mathcal{S}_i^c,\mathcal{S}_i}\hat{\Sigma}_{\mathcal{S}_i,\mathcal{S}_i}^{-1}||_\infty \leq 1 - \frac{1}{2}\eta^2.
$$

The proof of Lemma B.4 is then completed. $\qquad\square$

Denote $W_i = diag(\hat{\omega}_i)$. Applying the KKT condition, we get that

$$
-\frac{2}{n}\hat{\mathbf{\Upsilon}}_{-i}^T\mathbf{P}_i(\mathbf{P}_i\mathbf{Y}_i - \mathbf{P}_i\hat{\mathbf{\Upsilon}}_{-i}\hat{\boldsymbol{\beta}}_i) + \nu_i W_i \alpha_i = 0, \tag{34}
$$

where $\alpha_i \in \mathbb{R}^{Kp-K}$, satisfying $\alpha_{ij}\, I(\hat{\beta}_{ij} \neq 0) = sign(\hat{\beta}_{ij})$.

Using the equation $\mathbf{P}_i\mathbf{Y}_i = \mathbf{P}_i\mathbf{\Upsilon}_{-i}\boldsymbol{\beta}_i + \mathbf{P}_i\boldsymbol{\epsilon}_i$, we have that

$$
\begin{aligned}
&\hat{\mathbf{\Upsilon}}_{-i}^T\mathbf{P}_i(\mathbf{P}_i\mathbf{Y}_i - \mathbf{P}_i\hat{\mathbf{\Upsilon}}_{-i}\hat{\boldsymbol{\beta}}_i) \\
&= \hat{\mathbf{\Upsilon}}_{-i}^T\mathbf{P}_i[\mathbf{P}_i\boldsymbol{\epsilon}_i + \mathbf{P}_i(\mathbf{\Upsilon}_{-i} - \hat{\mathbf{\Upsilon}}_{-i})\hat{\boldsymbol{\beta}}_i) + \mathbf{P}_i\hat{\mathbf{\Upsilon}}_{-i}(\boldsymbol{\beta}_i - \hat{\boldsymbol{\beta}}_i)] \\
&= \underbrace{\hat{\mathbf{\Upsilon}}_{-i}^T\mathbf{P}_i(\boldsymbol{\epsilon}_i + (\mathbf{\Upsilon}_{-i} - \hat{\mathbf{\Upsilon}}_{-i})\hat{\boldsymbol{\beta}}_i)}_{T_{41}} - \underbrace{\hat{\mathbf{\Upsilon}}_{-i}^T\mathbf{P}_i\hat{\mathbf{\Upsilon}}_{-i}(\hat{\boldsymbol{\beta}}_i - \boldsymbol{\beta}_i)}_{T_{42}}.
\end{aligned}
\tag{35}
$$

With the definition of $\hat{\Sigma}_i$, (34) implies that,

$$
\frac{1}{n}T_{41} - \hat{\Sigma}_i(\hat{\boldsymbol{\beta}}_i - \boldsymbol{\beta}_i) = \frac{1}{2}\nu_i W_i \alpha_i, \tag{36}
$$

We consider the rows indexed by $\mathcal{S}_i$ and $\mathcal{S}_i^c$ in both sides of equation (36) which can be decomposed as

$$
\begin{cases}
\dfrac{1}{n}T_{41,\mathcal{S}_i} - \hat{\Sigma}_{\mathcal{S}_i,\mathcal{S}_i}(\hat{\boldsymbol{\beta}}_{\mathcal{S}_i} - \boldsymbol{\beta}_{\mathcal{S}_i}) = \dfrac{1}{2}\nu_i W_{\mathcal{S}_i}\alpha_{\mathcal{S}_i}, \\
\dfrac{1}{n}T_{41,\mathcal{S}_i^c} - \hat{\Sigma}_{\mathcal{S}_i^c,\mathcal{S}_i}(\hat{\boldsymbol{\beta}}_{\mathcal{S}_i} - \boldsymbol{\beta}_{\mathcal{S}_i}) = \dfrac{1}{2}\nu_i W_{\mathcal{S}_i^c}\alpha_{\mathcal{S}_i^c},
\end{cases}
\tag{37}
$$

where $T_{41,\mathcal{S}_i}$ and $T_{41,\mathcal{S}_i^c}$ denote the $\mathcal{S}_i$ and $\mathcal{S}_i^c$ rows for $T_{41}$, respectively.

From the first equation of (37), we can equate the error of estimation indexed by $\mathcal{S}_i$ as,

$$
\hat{\boldsymbol{\beta}}_{\mathcal{S}_i} - \boldsymbol{\beta}_{\mathcal{S}_i} = \hat{\Sigma}_{\mathcal{S}_i,\mathcal{S}_i}^{-1}(\frac{1}{n}T_{41,\mathcal{S}_i} - \frac{1}{2}\nu_i W_{\mathcal{S}_i}\alpha_{\mathcal{S}_i}). \tag{38}
$$

Scaling $\nu_i$ by $\frac{1}{2}\frac{4-\eta^2}{\eta^2}$ and using the same method in the proof of Lemma B.3, we have at least probability at least $1 - \delta_{min} - p\sum_{k=1}^{K}\delta^{(k)}$, such that

$$
||T_{41}||_\infty \leq \frac{1}{2}\frac{\eta^2}{4-\eta^2}n\nu_i||\hat{\boldsymbol{\omega}}_{\mathcal{S}_i}||_\infty.
$$

Therefore, we can bound the infinity norm of the above error as,

$$
\begin{aligned}
||\hat{\boldsymbol{\beta}}_{\mathcal{S}_i} - \boldsymbol{\beta}_{\mathcal{S}_i}||_\infty &\leq ||\hat{\Sigma}_{\mathcal{S}_i,\mathcal{S}_i}^{-1}||_\infty (\frac{1}{n}||T_{41}||_\infty + \frac{1}{2}\nu_i||\hat{\boldsymbol{\omega}}_{\mathcal{S}_i}||_\infty) \\
&\leq \frac{\eta+2}{2}\tau_i\frac{2}{4-\eta^2}\nu_i||\hat{\boldsymbol{\omega}}_{\mathcal{S}_i}||_\infty \\
&= \frac{\nu_i\tau_i||\hat{\boldsymbol{\omega}}_{\mathcal{S}_i}||_\infty}{2-\eta} \leq \min_{j\in\mathcal{S}_i}|\boldsymbol{\beta}_{ij}|.
\end{aligned}
$$

It indicates the largest absolute error is no larger than the minimal absolute signal, which leads to

$$
sign(\hat{\boldsymbol{\beta}}_{\mathcal{S}_i}) = sign(\boldsymbol{\beta}_{\mathcal{S}_i}).
$$

Next we will validate the second equation of (37) using the decomposition of error in (38), we have that

$$
\begin{aligned}
||\frac{1}{n}T_{41,\mathcal{S}_i^c} &- \hat{\Sigma}_{\mathcal{S}_i^c,\mathcal{S}_i}\hat{\Sigma}_{\mathcal{S}_i,\mathcal{S}_i}^{-1}(\frac{1}{n}T_{41,\mathcal{S}_i} - \frac{1}{2}\nu_i W_{\mathcal{S}_i}\alpha_{\mathcal{S}_i})||_\infty \\
&\leq \frac{1}{2}\frac{\eta^2}{4-\eta^2}\nu_i||\hat{\boldsymbol{\omega}}_{\mathcal{S}_i}||_\infty + ||\hat{\Sigma}_{\mathcal{S}_i^c,\mathcal{S}_i}\hat{\Sigma}_{\mathcal{S}_i,\mathcal{S}_i}^{-1}||_\infty(\frac{1}{2}\frac{\eta^2}{4-\eta^2}\nu_i||\hat{\boldsymbol{\omega}}_{\mathcal{S}_i}||_\infty + \frac{1}{2}\nu_i||\hat{\boldsymbol{\omega}}_{\mathcal{S}_i}||_\infty) \\
&\leq (\frac{1}{2}\frac{\eta^2}{4-\eta^2} + \frac{2-\eta^2}{2}\frac{2}{4-\eta^2})\nu_i||\hat{\boldsymbol{\omega}}_{\mathcal{S}_i}||_\infty = \frac{1}{2}\nu_i||\hat{\boldsymbol{\omega}}_{\mathcal{S}_i}||_\infty \\
&\leq \frac{1}{2}\nu_i||\hat{\boldsymbol{\omega}}_{\mathcal{S}_i^c}||_\infty.
\end{aligned}
$$

From the construction above, we have proved $sign(\hat{\boldsymbol{\beta}}_i) = sign(\boldsymbol{\beta}_i)$ and thus complete the proof of Theorem B.4.

# C    Proofs of the Theorems in the Main Text

## C.1    Proof of Theorem 4.1

Theorem 4.1 directly follows Theorem B.3 and B.4.

## C.2    Proof of Theorem 4.2

Denote $\boldsymbol{\zeta}_i^{(k)} = \mathbf{X}_{\mathcal{I}_i}^{(k)}\boldsymbol{\phi}_{\mathcal{I}_i}^{(k)} + \boldsymbol{\epsilon}_i^{(k)}$, then $\mathbf{Y}_i^{(k)} = \mathbf{Y}_{-i}^{(k)} + \boldsymbol{\zeta}_i^{(k)}$ according to (1). We have

$$
\begin{aligned}
|R_i^{2(k)} - R_{0i}^{2(k)}| &= \frac{|||\mathbf{Y}_{-i}^{(k)}\hat{\boldsymbol{\gamma}}_i^{(k)} - \mathbf{Y}_i^{(k)}||_2^2 - ||\mathbf{Y}_{-i}^{(k)}\boldsymbol{\gamma}_i^{(k)} - \mathbf{Y}_i^{(k)}||_2^2|}{||\mathbf{Y}_i^{(k)}||_2^2} \\
&= \frac{[\mathbf{Y}_{-i}^{(k)}(\hat{\boldsymbol{\gamma}}_i^{(k)} - \boldsymbol{\gamma}_i^{(k)})]^T[\mathbf{Y}_{-i}^{(k)}(\hat{\boldsymbol{\gamma}}_i^{(k)} - \boldsymbol{\gamma}_i^{(k)}) - 2\boldsymbol{\zeta}_i^{(k)}]}{||\mathbf{Y}_i^{(k)}||_2^2} \\
&= \frac{|||\mathbf{Y}_{-i}^{(k)}(\hat{\boldsymbol{\gamma}}_i^{(k)} - \boldsymbol{\gamma}_i^{(k)})||_2^2 - 2[\mathbf{Y}_{-i}^{(k)}(\hat{\boldsymbol{\gamma}}_i^{(k)} - \boldsymbol{\gamma}_i^{(k)})]^T\boldsymbol{\zeta}_i^{(k)}|}{||\mathbf{Y}_i^{(k)}||_2^2}.
\end{aligned} \tag{39}
$$

By Cauchy-Schwarz inequality, we have that

$$
[\mathbf{Y}_{-i}^{(k)}(\hat{\boldsymbol{\gamma}}_i^{(k)} - \boldsymbol{\gamma}_i^{(k)})]^T\boldsymbol{\zeta}_i^{(k)} \leq ||\mathbf{Y}_{-i}^{(k)}(\hat{\boldsymbol{\gamma}}_i^{(k)} - \boldsymbol{\gamma}_i^{(k)})||_2||\boldsymbol{\zeta}_i^{(k)}||_2, \tag{40}
$$

where

$$
||\boldsymbol{\zeta}_i^{(k)}||_2^2 \lesssim ||\mathbf{X}_{\mathcal{I}_i}^{(k)}\boldsymbol{\phi}_{\mathcal{I}_i}^{(k)}||_2||\boldsymbol{\epsilon}_i^{(k)}||_2 \leq \sqrt{n}||\boldsymbol{\phi}_{\mathcal{I}_i}^{(k)}||_2||\boldsymbol{\epsilon}_i^{(k)}||_2. \tag{41}
$$

Note that $||\boldsymbol{\epsilon}_i^{(k)}||_2^2$ follows the $\chi^2$ distribution, so we have that, with probability at least $1 - \delta_{min}$

$$
||\boldsymbol{\epsilon}_i^{(k)}||_2 \leq \sqrt{n^{(k)} + 2\sqrt{n^{(k)}log(\frac{1}{\delta_{min}})} + 2log(\frac{1}{\delta_{min}})}. \tag{42}
$$

Furthermore, we have $||\mathbf{Y}_i||_2^{2(k)} \asymp n^{(k)}$ due to normaliztion. Then collecting Theorem (4.1), equations (39), (40), (41), (42), we have that, with probability at least $1 - 2\delta_{min} - p\delta^{(k)}$,

$$
\begin{aligned}
|R_i^{2(k)} - R_{0i}^{2(k)}| \quad &\lesssim \quad \frac{|S_i|g_n\{d \vee \log(\frac{1}{\delta^{(k)}}) \vee ||\boldsymbol{\pi}_i||_2^{2(k)} \vee log\frac{d}{\delta_{min}}\}}{n^{(k)2}} \\
&+ \quad \frac{\sqrt{|S_i|g_n\{d \vee \log(\frac{1}{\delta^{(k)}}) \vee ||\boldsymbol{\pi}_i||_2^{2(k)} \vee log\frac{d}{\delta_{min}}\}}\sqrt{n^{(k)}}||\boldsymbol{\phi}_{\mathcal{I}_i}^{(k)}||_2 h_n}{n^{(k)}}
\end{aligned}
$$
(43)

where $h_n = \sqrt{n^{(k)} + 2\sqrt{n^{(k)}log(\frac{1}{\delta_{min}})} + 2log(\frac{1}{\delta_{min}})}$.

We can derive the same bound for the whole system, with probability at least $1 - p(2\delta_{min} + p\delta^{(k)})$, we have that

$$
\sum_{i=1}^{p}|R_i^{2(k)} - R_{0i}^{2(k)}|
$$
(44)

$$
\begin{aligned}
&\lesssim \quad \frac{|S_i|g_n\{d \vee \log(\frac{1}{\delta^{(k)}}) \vee ||\boldsymbol{\pi}_i||_2^{2(k)} \vee log\frac{d}{\delta_{min}}\}}{n^{(k)}} \\
&+\frac{\sqrt{|S_i|g_n\{d \vee \log(\frac{1}{\delta^{(k)}}) \vee ||\boldsymbol{\pi}_i||_2^{2(k)} \vee log\frac{d}{\delta_{min}}\}}\sqrt{n^{(k)}}||\boldsymbol{\phi}_{\mathcal{I}_i}^{(k)}||_2 h_n}{n^{(k)}}.
\end{aligned}
$$
(45)

In the following proofs, we will bound the error of the $C^2$ statistics.

$$
\begin{aligned}
&\sum_{j=1}^{p}|C_j^{2(k)} - C_{j0}^{2(k)}| \\
&= \quad \sum_{j=1}^{p}\sum_{i=1}^{p}\frac{|||\mathbf{Y}_j^{(k)}\hat{\boldsymbol{\gamma}}_{ij}^{(k)} - \mathbf{Y}_i^{(k)}||_2^2 - ||\mathbf{Y}_j^{(k)}\boldsymbol{\gamma}_{ij}^{(k)} - \mathbf{Y}_i^{(k)}||_2^2|}{||\mathbf{Y}_i^{(k)}||_2^2} \\
&= \quad \sum_{j=1}^{p}\sum_{i=1}^{p}\frac{|[\mathbf{Y}_j^{(k)}(\hat{\boldsymbol{\gamma}}_{ij}^{(k)} - \boldsymbol{\gamma}_{ij}^{(k)})]^T[\mathbf{Y}_j^{(k)}(\hat{\boldsymbol{\gamma}}_{ij}^{(k)} - \boldsymbol{\gamma}_{ij}^{(k)}) - 2\mathbf{Y}_i^{(k)} + 2\mathbf{Y}_j^{(k)}\boldsymbol{\gamma}_{ij}^{(k)})]|}{||\mathbf{Y}_i^{(k)}||_2^2}.
\end{aligned}
$$
(46)

Note that

$$
\begin{aligned}
&\sum_{j=1}^{p}\sum_{i=1}^{p}|[\mathbf{Y}_j^{(k)}(\hat{\boldsymbol{\gamma}}_{ij}^{(k)} - \boldsymbol{\gamma}_{ij}^{(k)})]^T[\mathbf{Y}_j^{(k)}(\hat{\boldsymbol{\gamma}}_{ij}^{(k)} - \boldsymbol{\gamma}_{ij}^{(k)}) - 2\mathbf{Y}_i^{(k)} + 2\mathbf{Y}_j^{(k)}\boldsymbol{\gamma}_{ij}^{(k)})]| \\
&= \quad \sum_{j=1}^{p}\sum_{i=1}^{p}|||\mathbf{Y}_j^{(k)}(\hat{\boldsymbol{\gamma}}_{ij}^{(k)} - \boldsymbol{\gamma}_{ij}^{(k)})||_2^2 - 2(\hat{\boldsymbol{\gamma}}_{ij}^{(k)} - \boldsymbol{\gamma}_{ij}^{(k)})\mathbf{Y}_j^{(k)T}\mathbf{Y}_i^{(k)} \\
&\quad +2(\hat{\boldsymbol{\gamma}}_{ij}^{(k)} - \boldsymbol{\gamma}_{ij}^{(k)})\boldsymbol{\gamma}_{ij}^{(k)}\mathbf{Y}_j^{(k)T}\mathbf{Y}_j^{(k)}|,
\end{aligned}
$$
(47)

and

$$
\mathbf{Y}_j^{(k)T}\mathbf{Y}_i^{(k)} \le ||\mathbf{Y}_j^{(k)}||_2||\mathbf{Y}_i^{(k)}||_2 \asymp n^{(k)}.
$$

Then with equation (28), we have

$$
\begin{aligned}
&\sum_{j=1}^{p}|C_j^{2(k)} - C_{j0}^{2(k)}| \\
&\lesssim \quad \sum_{j=1}^{p}\sum_{i=1}^{p}\frac{||\mathbf{Y}_j^{(k)}(\hat{\boldsymbol{\gamma}}_{ij}^{(k)} - \boldsymbol{\gamma}_{ij}^{(k)})||_2^2 + n^{(k)}|\hat{\boldsymbol{\gamma}}_{ij}^{(k)} - \boldsymbol{\gamma}_{ij}^{(k)}|(||B||_1 \vee 1)}{n^{(k)}} \\
&\lesssim \quad \sum_{i=1}^{p}\frac{\sum_{j=1}^{p}||\mathbf{Y}_j^{(k)}||_2^2||(\hat{\boldsymbol{\gamma}}_{ij}^{(k)} - \boldsymbol{\gamma}_{ij}^{(k)})||_2^2 + n^{(k)}||\hat{\boldsymbol{\gamma}}_i^{(k)} - \boldsymbol{\gamma}_i^{(k)}||_1(||B||_1 \vee 1)}{n^{(k)}} \\
&\lesssim \quad \sum_{i=1}^{p}\frac{n^{(k)}||\hat{\boldsymbol{\gamma}}_{ij}^{(k)} - \boldsymbol{\gamma}_{ij}^{(k)}||_2^2 + n^{(k)}\sqrt{|\mathcal{S}_i|}||\hat{\boldsymbol{\gamma}}_i^{(k)} - \boldsymbol{\gamma}_i^{(k)}||_2(||B||_1 \vee 1)}{n^{(k)}}.
\end{aligned}
$$
(48)

Applying theorem 4.1 in the whole system, we have that, with probability at least $1 - p(\delta_{min} + p\delta^{(k)})$

$$\sum_{j=1}^{p} |C_j^{2(k)} - C_{j0}^{2(k)}|$$

$$\lesssim \frac{|\mathcal{S}_i| g_n \{ d \vee \log(\frac{1}{\delta^{(k)}}) \vee ||\boldsymbol{\pi}_i||_2^{2(k)} \vee log \frac{d}{\delta_{min}} \}}{n^{(k)}}$$

$$+ \frac{\sqrt{n^{(k)} |\mathcal{S}_i|^2 g_n \{ d \vee \log(\frac{1}{\delta^{(k)}}) \vee ||\boldsymbol{\pi}_i||_2^{2(k)} \vee \log \frac{d}{\delta_{min}} \}}(||B||_1 \vee 1)}{n^{(k)}}. \qquad (49)$$

Equation (44, 49) lead to the discussion in theorem 4.2 and the equations simplifies to theorem 4.2.

## D    Details in the Simulation Study

In Figure 6, we present one plot of the three networks used for the simulation study that corresponds to Figure 4, showing the causal relations in the baseline network DCG III in black, the deviated causal effects DCG I vs. DCG III, DCG II vs. DCG III, and DCG I vs. DCG II in blue, red, and green, respectively.

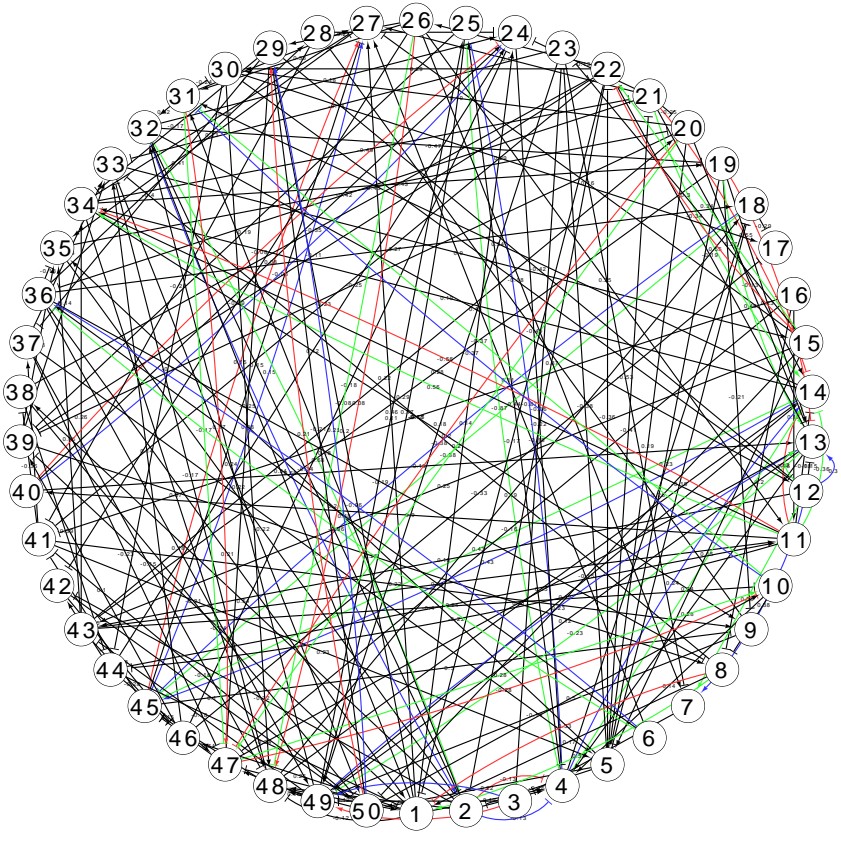

Figure 6: Plot of all causal effects among three networks used for the simulation study with baseline DCG III, DCG I vs. DCG III, DCG II vs. DCG III, and DCG I vs. DCG II in black, blue, red, and green, respectively.

## E    Limitations

Although we have developed a limited-information likelihood method to avoid optimizing too many model parameters as the full-information likelihood method does, the proposed method may still be

challenged by large $K$ and massive total sample size $n$ when there are too many cohorts to compare. When $K$ is too large, each task in the algorithm (identifying and estimating causal effects for a single responder) has to estimate $K(p-1)$ parameters with an $n \times (K(p-1))$ design matrix, possibly demanding a large amount of memory.

We have developed our algorithm for the case to compare all other networks to a single baseline network and provide theoretical analysis. In practice, we may be interested in the deviated effects of each network from the average effects. Our algorithm can be adopted for such a case. However, it is challenging to develop an appropriate theoretical analysis for this case, and deserves further study.