# OpenReview forum: "Analysis of Variance of Multiple Causal Networks"
_NeurIPS.cc/2023/Conference — NeurIPS 2023 poster_

### Official Review · Reviewer_Daft · 2023-07-06

**Soundness:** 3 good
**Presentation:** 4 excellent
**Contribution:** 3 good
**Rating:** 7
**Confidence:** 2

**Summary:**

This paper proposes a single structural model to simultaneously construct multiple networks so as to identify causalities varying across multiple cohorts while identifying stable ones.  Each causal network is represented via a directed cyclic graph, and the authors propose an analysis of variance (ANOVA) algorithm (NetANOVA) to identify causalities that are different across networks and an SVD based correspondence analysis to identify the key drivers and responders. Theoretical properties of the algorithm for DCG construction are established and simulations and a data example illustrate the working of their methodology.


**Strengths:**

The paper deals with an important problem and appears to be a novel contribution as the problem of investigating causality with multiple networks as done here, has not been studied in the literature. It is presented in a very clear manner and is easy to read in most parts. Some technical aspects could be clarified further (as mentioned under Questions below). The simplicity of their framework; theoretical guarantees and computational scalability are the main strengths in my opinion.

**Weaknesses:**

The svd based technique employed for correspondence analysis to detect key responder drivers pairs, is not clear. It would be helpful if the authors could elaborate further on this while also providing the intuition behind this. Bootstrapping is used to achieve this- what sort of a bootstrap method is used and why?
Intuitively, I would expect the accuracy of inferred DCGs  to depend on the range of cohort sizes (min of n^k's to max of n^k's) with lower accuracy as the range increases. Is this the case? If yes, is it evident from the theoretical results?

**Questions:**

Please answer questions mentioned under weaknesses above.

Section 5: why is error variance of 0.1^2 a good choice? Can you provide the corresponding signal-to-noise ratio?
Section 3.2: Would it help in interpretation to normalize the coefficient of cause as well so that it lies in (0,1] as R^2?

Please check the paper for typos (e.g. p8, line 257: n_2, n_3 instead?; p5, line 156: responders; p2, line 54: endogenous etc).

**Limitations:**

I did not find a discussion of limitations of the method. Please include a brief discussion of these in the paper.

---

> ### Author Rebuttal · Authors · 2023-08-10
>
> “The svd based technique employed for correspondence analysis to detect key responder drivers pairs, is not clear. It would be helpful if the authors could elaborate further on this while also providing the intuition behind this. Bootstrapping is used to achieve this- what sort of a bootstrap method is used and why?”
>
> It is intimidating to capture key information on vast information from multiple large networks. So we have proposed the correspondence analysis to reveal important clusters of drivers, responders, or even driver-responder pairs for their similarity/dissimilarity across causal networks. For the possible causal effect of node j on node i, its difference between cohort k and cohort l is estimated as hat{γ}_{ij}^{(k)}-hat{γ} _{ij}^{(l)}, however, this value may be data-sensitive and not stable. Therefore, we take advantage of bootstrap results to estimate its standard deviation and further standardize the difference to construct the matrix Z^{(k,l)} for SVD. With the top singular vectors catching the major variations cross responders/drivers, the correspondence analysis will reveal important clusters.
>
> For bootstrap, we took nonparametric bootstrap by randomly sampling the data with replacement, because it is difficult to develop asymptotic distributions on the estimated parameters for the high-dimensional networks.
>
> “Intuitively, I would expect the accuracy of inferred DCGs to depend on the range of cohort sizes (min of n^k's to max of n^k's) with lower accuracy as the range increases. Is this the case? If yes, is it evident from the theoretical results?”
>
> It is a nice point to make. Yes, with fixed K cohorts, the smallest cohort size, which is denoted as n_min, and the sum of all the cohort sizes, which is denoted as n, determine the accuracy of inferred networks and deviated causal effects.  As shown in Theorem 4.1, the error bound is inversely proportional to n and is also proportionally to g_n which, as shown in Line 427, is inversely proportional to sqrt{n_min}. In summary, when n_min or n decreases, we have a larger error bound.
>
> “Section 5: why is error variance of 0.1^2 a good choice? Can you provide the corresponding signal-to-noise ratio? Section 3.2: Would it help in interpretation to normalize the coefficient of cause as well so that it lies in (0,1] as R^2?”
>
> We chose the error variance of 0.1^2 following works on single causal networks [4, 13]. With respect to the signal-to-noise ratio (defined as var(γ_j Y_j )/var(Y_i |Y_{-i}) when node i’s value Y_i has causal contribution γ_j Y_j  from node j), it ranges from .02861 to 9.8534 with average at .7559 and median at .2335 (based on one simulated dataset).
>
> It is an interesting point on normalizing the coefficient of cause into [0,1]. One way is to calculate the averaged coefficient of cause via dividing it by the number of responders taking this driver. Alternatively, we may divide the total driver’s causal contributions in terms of variance by the sum of variances of all responders. Both measures should be read together with the total number of responders as a driver with too many responders may be of high interest even if the causal contribution to each responder is low.
>
> “Please check the paper for typos (e.g. p8, line 257: n_2, n_3 instead?; p5, line 156: responders; p2, line 54: endogenous etc).”
>
> Thanks for pointing that out. We will correct the typos in our revised version.
>
> “I did not find a discussion of limitations of the method. Please include a brief discussion of these in the paper.”
>
> Although we have developed a limited-information likelihood method to avoid optimizing too many model parameters as the full-information likelihood method does, the proposed method may still be challenged by large K and massive total sample size n when there are too many cohorts to compare. When K is too large, each task in the algorithm (identifying and estimating causal effects for a single responder) has to estimate K(p-1) parameters with an n×(K(p-1)) design matrix, possibly demanding a large amount of memory.
>
> We have developed our algorithm for the case to compare all other networks to a single baseline network and provide theoretical analysis. In practice, we may be interested in the deviated effects of each network from the average effects. Our algorithm can be adopted for such a case. However, it is challenging to develop an appropriate theoretical analysis for this case, and deserves further study.

---

### Official Review · Reviewer_Wm1e · 2023-07-07

**Soundness:** 3 good
**Presentation:** 2 fair
**Contribution:** 2 fair
**Rating:** 6
**Confidence:** 2

**Summary:**

In this paper, the authors proposed NetANOVA, an algorithm that simultaneously constructs multiple causal networks and infer their disparities. Theoretical justification of the proposed method is also derived. The paper further proposes measures for variable’s contribution to receivers and responders. Overall, the problem seems solid and experiment results reveal its effectiveness.

**Strengths:**

The task this paper is tackling is of interest and importance. The proposed method is unified on handling multiple causal networks simultaneously and is scalable in various aspects.

The derivation of the method is clear and reasonable. The authors discuss annotations and assumptions of their proposed method.  They show theoretical justification of the algorithm, which consolidates their work.


**Weaknesses:**

The paper is not really good to read. Section 2 starts deriving the method directly without further explanation and formulation of the problem. The authors should consider adding a subsection before the method to formally describe the problem.

The experiments seem good. However, no related methods are compared. The authors  mentioned several related works in the introduction and they highlighted advantages of the proposed method (scalability in various aspects). The experiments do not discuss how their methods compare with others.


**Questions:**

How is the proposed method compared with related works?

The model is derived on the basis of several assumptions. While in experiments, do those assumptions really hold on the datasets?

Do the authors conduct runtime analysis of their proposed method?

**Limitations:**

It will be good if the authors can provide a formal definition on the problem they are trying to solve.

---

> ### Author Rebuttal · Authors · 2023-08-10
>
> “The paper is not really good to read. Section 2 starts deriving the method directly without further explanation and formulation of the problem. The authors should consider adding a subsection before the method to formally describe the problem.”
>
> The main purpose is to construct and compare multiple causal networks to gain knowledge of cross-cohort changes in causal effects as well as commonly shared ones (with a focus on deviated causal effects of other networks from a baseline). We are sorry that we have to make the description concise to include all aspects of the algorithm development as well as the theoretical/empirical evaluation. We have a rather longer version of the manuscript and would be happy to make it available in arXiv.
>
> “The model is derived on the basis of several assumptions. While in experiments, do those assumptions really hold on the datasets?”
>
> All networks are simulated according to Assumption 1. Assumptions taken for theoretical works are mostly weak and widely adopted in the literature. Specifically, in our experiment, both Assumptions B.1 and B.2 put assumptions on the spectrums of the variance and covariance matrix as well as the IV size by Assumption B,1,  and are satisfied by our simulation settings. Assumption B.3 holds because it is about augmenting the design matrix but the order of the eigenvalues remains unchanged in our simulation.  Assumption B.4 requires limited association between drivers and non-drivers, which also holds in our case.
>
> “Do the authors conduct runtime analysis of their proposed method?”
>
> For our simulation study with one core on Rome CPU @ 2.0 GHz, analysis of one simulated dataset (without parallel computation) took 11,001s,  23,508s,  and 47,883s for sample sizes 200, 500, and 1000 respectively. The use of one node with 128 cores in HPC would cut the time to 86s, 183s, and 374s respectively, taking advantage of the parallel nature of the algorithm.

---

> > ### Comment · Reviewer_Wm1e · 2023-08-21
> >
> > Thanks for the authors' review. I would like to keep my current raing.

---

### Official Review · Reviewer_bLo4 · 2023-07-09

**Soundness:** 3 good
**Presentation:** 3 good
**Contribution:** 2 fair
**Rating:** 5
**Confidence:** 2

**Summary:**

The paper introduces NetANOVA, an algorithm designed for parallel computation to construct a unified structural model for multiple causal networks, or DCFs. NetANOVA utilizes analysis of variance (ANOVA) to identify causalities that differ across networks, as well as important drivers and responders. It is scalable to large data sizes, computational environments, and model complexities, allowing for efficient analysis of multiple networks.

**Strengths:**

1. The paper introduces a two-stage parallelizable algorithm that scales when multiple cohorts exist.
2. The paper gives solid theoretical results on the consistency for coefficients of determination and cause.

**Weaknesses:**

1. Experimental results on real data is limited to three DCGs, and it does not contain comparisons with baseline approaches.
2. The theoretical result does not include the computation complexity of NetANOVA.

**Questions:**

1. In NetANOVA, most computations are matrix products and inversions. Can you briefly estimate the computation cost required when n and k scales?
2. In Figure 4, are DCG(I/II/III) curated by humans? If so, what is the cost of curating each DCG?

**Limitations:**

See "weaknesses."

---

> ### Author Rebuttal · Authors · 2023-08-10
>
> “In NetANOVA, most computations are matrix products and inversions. Can you briefly estimate the computation cost required when n and k scales?”
>
> Assuming bar{n}=sum_{k=1}^K n^{(k)} the average sample size,  we can break down the computational complexity as follows: The complexity associated with ISIS is O(K bar{n} p), complexity with ridge regression is O(K bar{n}^3 ), projection’s complexity is O(K bar{n}^3 ), complexity introduced by adaptive lasso is O(K^2 bar{n} p). Collectively, the computational cost for each node is O(K^2 bar{n} p) + O(K bar{n}^3).
>
> “In Figure 4, are DCG(I/II/III) curated by humans? If so, what is the cost of curating each DCG?”
>
> For each sample size (200, 500, or 1000), we have randomly simulated 100 sets of networks (with each set including 3 DCGs) as described in Lines 229-236,  and accordingly simulated one dataset from each set. Figure 4.a shows a part of one simulated set with a sample size 500 (the whole network is shown in Figure 6). We applied our algorithm to the data simulated from this set and conducted correspondence analysis of the results.  Figure 4.b-d show the plot of the correspondence analysis. Figure 4.a was obtained via the freely available software Cytoscape. and Figure 4.b-d was from R with packages ggplot and ggrepel.

---

### Official Review · Reviewer_ufKZ · 2023-07-10

**Soundness:** 3 good
**Presentation:** 2 fair
**Contribution:** 3 good
**Rating:** 5
**Confidence:** 4

**Summary:**

This paper presents a unified structural model that describes multiple DCGs in one model and develops a limited-information-based method to simultaneously infer networks and their disparities. Furthermore, it provides robust non-asymptotic theoretical properties. And it is applied to synthetic and real datasets to show its performance.


**Strengths:**

The paper proposes a new model that manages to describes multiple DCGs within a single model, and further performs a comprehensive study on it, including algorithmic development, theoretical analysis and synthetic and real-data experiments.


**Weaknesses:**

The paper proposes such a model that describes multiple DCGs by stacking K-network-specific structural models in matrix forms with one network chosen as the baseline. However, we fail to see the benefits of proposing such a "unified" model, nor any mathematical novelty in model construction. In the problem setup, there is no relation between or information shared by K networks. And in algorithm development, the algorithm is almost like dealing with network-specific parameters independently. In experiments, we did not see any benefits this approach manifests.

Due to this straightforward way of construction, it can be foreseen that the algorithmic development and theoretical analyses naturally follow without particular technical issues. For algorithm development, it seems more or less doing network-specific inference and then stacking results. The theoretical results are mainly these from high-dimensional statistics (specifically l1-related theory on variable screening and selection). Considering the almost independent treatment to each network in algorithm, the theoretical results naturally follow by repeating these theories from analysis of structural models; thus, the introduction of these high-dim stats tools may not be highly appreciated.

The simulation study and real-data applications only show the "success" application of the proposed model. Without the basic comparison to the state-of-the-art or the independent treatments to each network using classical structural model methods, we have no idea on how good the performance of the proposed method is and what additional benefits we gain by modelling multiple networks via a single model.

The literature review in Introduction is not good, which takes less than 1/3 space of Introduction. It is not a critical review, and it basically lists 6 papers (line 24-38) and tells what they did. These work are neither being explained on their relations to this paper nor commented on their strengths/weakness/debates. Just a list of work, which however is not sufficient to cover important work that closely relates to methods and theories in this paper. What's worse, the sentence at lien 25-26 reads like something irrelevant copied from other biological papers.


**Questions:**

See the weakness section. We do not have any further technical questions.

**Limitations:**

We do not see any potential negative societal impact of this paper.

---

> ### Author Rebuttal · Authors · 2023-08-10
>
> “… fail to see the benefits of proposing such a "unified" model, nor any mathematical novelty in model construction. … no relation between or information shared by K networks.”
>
> We disagree with the reviewer’s claims on the benefit and novelty. To the best of our knowledge, we are the first to study and compare multiple causal networks, which allows us to address two types of problems: i) Deviation of other networks to a single baseline (e.g., network of the normal/healthy population); ii) Variation (i.e., difference from each other) across multiple networks.
>
> Our method is novel in its strategically unifying multiple causal networks, shown from Line 86 to Line 95, which allows developing the algorithm NetANOVA to directly identify and estimate deviated causal effects when compared to a single baseline. We are also the first to explicitly define coefficients of determination and coefficients of cause for causal networks. Our proposed correspondence analysis of causal effects across multiple networks is also new with its defined summary statistics which take advantage of parallel computation to obtain bootstrap results.
>
> The theoretical benefit of using a unified model is further demonstrated in theorem 4.1. As shown in Theorem 4.1, the error bound of each hat{β}\_i is related to n=sum_{k=1}^K n^{(k)}  due to the deal with the unified model. Instead, an independent construction of the k-th network will result in an error bound related to n^{(k)}.
>
> Please be aware that, i) the reparametrization in (3) requires a pooled model for all involved networks; ii) with K-th network as the baseline, the parameters in β\_i^{(k)} show the difference between k-th network and the baseline; iii) although the algorithm allows heterogeneous instrumental variables across different networks and therefore may conduct the first stage independently, the second stage has to identify and estimate the deviated causal effects in hat{β}_i which are defined across networks. That is, at this key step, the unified model is directly constructed at Stage 2.3 of Algorithm 1. More details on the second stage can be found under “The inference stage” starting at Line 140.
>
> “… the algorithmic development and theoretical analyses naturally follow without particular technical issues. … thus, the introduction of these high-dim stats tools may not be highly appreciated.”
>
> We disagree with the reviewer’s claim on the theoretical analysis (please refer to our above rebuttal on algorithmic development). For our first-ever algorithm that directly detects and compares multiple causal networks, the error bounds of the corresponding parameters, especially  {hat{β}\_i, i=1,2,...,p}, is not a straightforward extension of the results from individual networks. As shown in Theorem 4.1, the error bound of each hat{β}\_i is related to n=sum_{k=1}^K n^{(k)} due to the deal with the unified model. Instead, an independent construction of the k-th network will result in an error bound related to n^{(k)}.
>
> Technically, theoretical analysis of causal networks is challenged by inherent endogeneity and henceforth developed two-stage algorithms. Unifying multiple networks to enable direct comparison of multiple networks indeed makes it even more challenging in controlling error bounds while taking advantage of datasets from multiple populations.
>
> We focus our research on studying the deviation of other networks to a single baseline (e.g., network of the normal/healthy population), the first type which is popular in disease studies, especially cancer studies. As we stated in Line 89, our algorithm can be extended to study variation (i.e., differences from each other) across multiple networks. However, the parameters of a possibly unified model are involved with parameters from all networks, making it much more difficult (so further efforts are demanded) to develop an appropriate theoretical analysis.
>
> “The literature review in Introduction is not good, …. It is not a critical review, and it basically lists 6 papers (line 24-38) and tells what they did.”
>
> As we are presenting a first-ever algorithm for constructing and analyzing multiple causal networks based on structural models, we focus on our review of related work. The three methods (Liu et al., Cai et al., and Che et al.) are the major development along this direction but for a single network.
>
> “These work are neither being explained on their relations to this paper nor commented …”
>
> We disagree with the reviewer because we described the major idea of each method with comments on its advantages/disadvantages.
> “What's worse, the sentence at lien 25-26 reads like something irrelevant copied from other biological papers.”
>
> We again disagree with the reviewer. We have this sentence here to emphasize that constructing structural models with instrumental variables is mainly developed in the field of bioinformatics because natural instrumental variables are available in studying gene regulatory networks. This sentence leads to the review of the developed methods for structural models in bioinformatics.

---

> > ### Comment · Reviewer_ufKZ · 2023-08-16
> >
> > We thank the authors a lot for addressing carefully on the technical points we have missed. Indeed, after diving into your analysis details together with your responses, we agree that we largely underestimate your contributions. Some technical points are quite subtle, which led us to overlook the contribution of introducing an "unified model" (What we thought can be equivalently done by separately analyzing each network is actually not working or fails to hold competitive performance in theory).
> >
> > Thanks for your efforts on explaining the details. We have updated our grading.

---

> > > ### Author Response · Authors · 2023-08-20
> > >
> > > Thank you for updating your grading. Please let us know if you have any concerns.

---

### Official Review · Reviewer_mopu · 2023-07-13

**Soundness:** 2 fair
**Presentation:** 2 fair
**Contribution:** 2 fair
**Rating:** 4
**Confidence:** 3

**Summary:**

This paper proposes an algorithm, called NetANOVA, for constructing a unified structural model for multiple causal networks with cycles. The algorithm is designed for parallel computation and is scalable to data size and network complexity. It is able to infer causal networks beyond directed acyclic graphs (DAGs), such as directed cyclic graphs (DCGs), which are demanded to depict gene regulatory networks. The paper provides theoretical justification for the algorithm and demonstrates its feasibility and promise with a large-scale simulation study and a real data analysis to compare gene regulatory networks in healthy lung tissues and lung tissues with two types of cancer.

**Strengths:**

* It proposes a method to unify multiple cyclic graphs with a single structural model and also accommodate their disparities, which is a promising method for analyzing gene regulatory networks where feedback loops are regularly encountered.
* The algorithm is designed for parallel computation and is scalable to data size and network complexity.

**Weaknesses:**

* Instrumental variables-based identification of causal effect are proposed for DAGs. There is a lack of discussion on how this extends to causal structures that contains cycles. It is unclear to me if the identification results in 2.3 is correct when cycles exists.


**Questions:**

* The word "variational" is used frequently. However, this word usually refers to well-known methods/concepts in machine learning, such as variational inference. Is this word choice for "variational causality" validated?
* Regarding the weakness, how would the existence of cycles affect identification of causal effects using IV? Since most results on IV are either discussed under the potential outcome framework, or discussed using do-calculus when the underlying causal structure can be described by DAGs.


**Limitations:**

The authors do not explicitly discuss the limitations of this work.

---

> ### Author Rebuttal · Authors · 2023-08-10
>
> “Instrumental variables-based identification of causal effect are proposed for DAGs. There is a lack of discussion on how this extends to causal structures that contains cycles. It is unclear to me if the identification results in 2.3 is correct when cycles exists.”
>
> We agree that IV-based methods are mainly proposed for DAGs. However, the econometrics field has observed much development in systems of equations, including model identification, under the name "simultaneous equation models [20]. Our specification of model identification is adopted from the rank condition developed for “simultaneous equation models” but in terms of instrumental variables. A rather recent work on identification can be found in the work by Matzkin (2008).
>
> Matzkin, Rosa L. "Identification in nonparametric simultaneous equations models." Econometrica 76.5 (2008): 945-978.
>
> “The word "variational" is used frequently. However, this word usually refers to well-known methods/concepts in machine learning, such as variational inference. Is this word choice for "variational causality" validated?”
>
> Thanks for highlighting the potential ambiguity. In our manuscript, we employed the term “variational” to refer to variation/deviation across multiple networks. To avoid any confusion and ensure clarity, we will replace “variational” with “perturbational” in the revised version of our paper.

---

> > ### Comment · Reviewer_mopu · 2023-08-14
> >
> > Thank the authors for replying to the comments. For the second point, I think your response is fine. For the first point (identification), can you please be more specific as the reference mentioned in the rebuttal is not in the original manuscript?

---

> > > ### Author Response · Authors · 2023-08-20
> > >
> > > The system of equations is classically discussed in econometrics and is known as "simultaneous equation model” [20]. Furthermore, the systems of equations under “simultaneous equation models” [20] cover directed cyclic graphs (DCGs) as well as DAGs. The rank condition is a well-known yet fundamental identification result in econometrics which can be found in every textbook of econometrics wherever it introduces simultaneous equation models.
> > >
> > > The rank condition is necessary and sufficient for a system of equations to be identifiable. Our Assumption 1 basically states that any driver in a causal system should have its unique instrumental variable(s), which makes the rank condition hold. Our mention of the new reference by Matzkin (2008) only intends to point to a recent study in identification (we don’t need this reference to claim the identification of our models). Indeed, the work by Matzkin (2008) provides the identification of more general nonparametric models (which certainly cover the models in our paper).
> > >
> > > We consider that Assumption 1 leading to the rank condition is obvious, so we don’t think it is necessary to further detail in the text. Per the reviewer’s inquiry, we would like to illustrate more on this point here to consider a system with p endogenous variables in Y and q exogenous variables in X, which is described by YΓ+XB=ε. Without loss of generality, we consider the first equation for the first endogenous variable Y_1 as Y_1+Y_{R_1}γ_1+X_{S_1}β_1= ε_1. Note that R_1 includes the indices of all drivers of Y_1 and  S_1 includes the indices of all instrumental variables unique for Y_1. Accordingly, we can specify the other p-1 equations together as Y_1 Γ_1+Y_{R_1} Γ_2+Y_{-[1 R_1]} Γ_3+X_{S_1} B_1+X_{-S_1} B_2=ε. The rank condition states that, the first equation for Y_1 is identifiable if and only rank([Γ_3,B_2])=p-1. Apparently, Assumption 1 suggests that rank([Γ_3,B_2])=p-1 because any driver of Y_1 suggests nonzero components in its own row of B_2 (and zeros in other rows of B_2), and non-drivers of Y_1 are categorized into two groups: those with each having its own instrumental variables (so having nonzero components in its own row of B_2 but zeros in other rows of B_2), and those without any instrumental variables so not driver of any other endogenous variable (so have nonzero components in its own row of Γ_3 but zero components in other rows of Γ_3).

---

### Author Rebuttal · Authors · 2023-08-10

We sincerely appreciate all the reviewers for providing constructive comments, which provide us a chance to clarify some confusions and improve the quality of our work. We have carefully been through each comment and done our best to address each. While we have addressed each reviewer’s points in separate rebuttals, here we would like to address a common concern on “Lack of comparison to the state-of-the-art or the independent treatments to each network using classical structural model methods”.

To the best of our knowledge, our algorithm is the first one developed for multiple causal network analysis and there is no other state-of-the-art method available to compare to.

Pooling independently constructed networks seems appealing in terms of computation. However, as shown in Theorem 4.1, the error bound of each hat{β}\_i from our algorithm is inversely proportional to n=sum_{k=1}^K n^{(k)}. However, an independent construction of the k-th network will result in an error bound related to n^{(k)}. Furthermore, due to the high-dimensional property, the estimates of many parameters may follow mixture distributions. That is, unlike classical low-dimensional problems presenting asymptotic normal distributions, the estimated values of bootstrapped high-dimensional datasets may be mixed with many zeros and nonzero values. Therefore, even in the case that the computational challenge overtakes statistical efficiency so we have to pool independently constructed networks, it demands further development of appropriate strategies.

Because of no state-of-the-art method to compare and difficulty in pooling results from independently constructed networks, we focus our simulation studies on evaluating the feasibility of our algorithm, and its efficacy over different sample sizes.

---

### Decision · Program_Chairs · 2023-09-21

**Decision:**

Accept (poster)

**Comment:**

This paper proposes a methodology for estimating causal effects simultaneously in multiple directed cyclic graphs / structural models that are related, e.g., a baseline model and a few changed models. By pooling the data together, the paper shows that better estimation (with gains that depend on just how much the models are related.) Reviewers generally agree that this contribution is worth sharing with the community. However, several changes would significantly improve the paper. First, presentation needs to be tweaked. For instance, explain the model without equations first, and then instantiate it and, for the theory, include all key assumptions as part of the main text, so that the scope of when guarantees hold is clear. Second, the experiments currently do not have a baseline method to compare against. The authors are strongly encouraged to include one, e.g., perhaps even a method that estimates causal effects individually in each case. Ideally, this should be done along with highlighting the statistical and computational gains that the new methodology provides. Lastly, the authors are encouraged to incorporate all the minor suggestions provided by the reviewers, e.g., changing the choice of the word "variational", and better explaining the bootstrapping methodology and its dependence on the cohort sizes.